# LONG-FORM HALLUCINATION DETECTION WITH SELF-ELICITATION

## ABSTRACT

While Large Language Models (LLMs) have exhibited impressive performance in long-form question-answering tasks, they frequently present a hazard of producing factual inaccuracies or hallucinations. An effective strategy to mitigate this hazard is to leverage off-the-shelf LLMs to detect hallucinations after the generation. The primary challenge resides in the comprehensive elicitation of the intrinsic knowledge acquired during their pre-training phase. However, existing methods that employ complex reasoning chains predominantly fall short of addressing this issue. Moreover, since existing methods for hallucination detection tend to decompose the text into isolated statements, they are unable to understand the inherent in-context semantics in long-form content. In this paper, we propose a novel framework, SelfElicit, which synergizes the self-elicitation of intrinsic knowledge of large language models and long-form continuity understanding. Specifically, we leverage self-generated thoughts derived from prior statements as catalysts to elicit the expression of intrinsic knowledge, which is integrated with graph structures to alleviate induced hallucinations and guide the factual evaluation by effectively organizing the elicited knowledge. Extensive experiments on real-world QA datasets demonstrate the effectiveness of self-elicitation and the superiority of our proposed method.

## 1 INTRODUCTION

Large Language Models (LLMs) pre-trained on massive text corpora and fine-tuned to follow human instructions have shown remarkable performance in various neutral language tasks (Bai et al., 2023; Touvron et al., 2023; GLM et al., 2024). However, there remains a concern regarding their tendency to generate hallucinations (Bang et al., 2023), producing sentences with plausible looking yet factually unsupported content (Huang et al., 2023)[1] and hurting their reliability in real-world scenarios expecting factually-accurate responses (Wei et al., 2024). For example, a model-generated non-factual statement *"Gliclazide can be taken at any time of the day×, regardless of whether it is on an empty stomach or after meals×"* might mislead patients into taking medication at incorrect times since this medication is recommended to be taken with the meal (NHS, 2024). An important strategy to alleviate hallucinations is to detect hallucinations after the generation (Lee et al., 2023; Manakul et al., 2023; Mishra et al., 2024; Guan et al., 2024).

Numerous methods have been proposed for the hallucination detection task. Several methods rely on retrieval (Min et al., 2023; Xia et al., 2024; Li et al., 2023b; Wei et al., 2024; Yue et al., 2024; Sansford et al., 2024) or probes (Li et al., 2023a; Zhang et al., 2024a; Chuang et al., 2024; Wang et al., 2024), but external databases or probe training corpus are not always available in all scenarios. Therefore, many studies focus on using the intrinsic capabilities of off-the-shelf LLMs acquired through pre-training, where the key challenge is *how to effectively elicit the intrinsic knowledge from the models*. Some methods prompt to model to implicitly utilize their knowledge to identify hallucinations by assessing confidence levels (Kadavath et al., 2022; Mahaut et al., 2024; Zhao et al., 2024). In contrast, other methods explicitly elicit intrinsic knowledge to enhance detection accuracy. For example, some works (Manakul et al., 2023; Mündler et al., 2024; Miao et al., 2024) prompt the model to generate statements from various perspectives and contrast these statements to quantify

---

[1] In this paper, we mainly focus on factuality (external) hallucinations and leave faithfulness (internal) hallucinations for future work (Huang et al., 2023).

Figure 1: Schematic illustration of hallucination detection from long-form content. **(a)** Statements are isolatedly evaluated. **(b)** Prior statements are incorporated as context. Our method investigates and demonstrates **(c)** how prior self-generated thoughts can elicit models' intrinsic knowledge.

the semantic consistency. Other works (Kang et al., 2023; Dhuliawala et al., 2024; Farquhar et al., 2024; Setty & Setty, 2024) ask the model to answer verification questions generated according to facts within the statements. While insightful, we contend that these methods either require complex manual prompts or involve intricate reasoning processes, which limit their elicitation capacity and increase the risk of accumulated inaccuracies and hallucinations.

Additionally, an inherent characteristic of long-form content is the ==in-context semantics== *among sentences, a logical and consistent relationship between different elements of meaning*, such as coherence, comparison, and causality. For instance, the preceding statement, *"Gliclazide is an oral hypoglycemic medication"* and the subsequent statement, *"It is suitable for adult type 2 diabetes patients whose blood sugar cannot be adequately controlled by diet alone"* demonstrate logical coherence and progression. The first statement identifies the category and function of the medication while the second statement further elaborates on its medical application. However, existing long-form hallucination detection methods (Zhang et al., 2020; Min et al., 2023; Wei et al., 2024; Li et al., 2024a) generally decompose the long-form text into isolated statements that each is fact-checked individually (Figure 1 (a)), overlooking such semantic continuity and limiting their reasoning capabilities. Providing prior contextual information to models (Figure 1 (b)) can present a more natural chain of meanings, thereby benefiting both the understanding and evaluation of subsequent statements.

In this work, we present **SelfElicit**, an integrated framework designed to effectively elicit a model's intrinsic knowledge and utilize semantic continuity to improve hallucination detection in long-form content. Specifically, it follows an iterative process in which the model first evaluates the factuality of statements conditioned on prior contextual information. It then engages in reflection to elicit the intrinsic knowledge and finally incorporates these reflections as context to enhance subsequent evaluations (Figure 1 (c)). To mitigate hallucinations arising during the self-elicit process, we integrate a knowledge hypergraph into the iterative framework, which facilitates knowledge retention, deduplication, and resolution of inconsistencies. Our extensive experiments demonstrate that self-eliciting can act as an effective catalyst to improve both the factuality and diversity of models' knowledge expression and our method outperforms existing methods for long-form hallucination detection. To sum up, our contributions include:

- We study a novel concept of *self-eliciting* large language models for hallucination detection. We show that using self-generated thoughts from prior statements as catalysts prompts the models to effectively express intrinsic knowledge and facilitates hallucination detection.

- We propose a new framework, SelfElicit, for long-form hallucination detection, which synergizes the self-eliciting mechanism with semantic continuity understanding. We design a knowledge hypergraph to carefully organize the elicited knowledge and effectively alleviate hallucination snowballing.

- SelfElicit framework consistently demonstrates superior performance in long-form hallucination detection using real-world datasets with modern language models. We further show that self-eliciting enhances knowledge expression with better factuality and diversity.

## 2 PRELIMINARIES

### 2.1 TASK

In this paper, we investigate the task of retrieval-free long-form hallucination detection. Given the user query and the original long-form response generated by a generator LM, the target is to utilize

an analyzer LM to evaluate whether there is any factual incorrectness in the response. This task focuses on hallucination detection in the post-generation phase and uses the intrinsic capabilities of off-the-shelf LLMs, rather than relying on external databases or fine-tuning.

**Long-form Hallucination Detection**. Given a user query $Q$ and an original response $R$ that can be parsed into sentences $R = \{r_1, r_2, \cdots\}$, the long-form hallucination detection task is to classify whether there is any factual incorrectness in each sentence and the entire response. Formally,

$$\hat{y}_1, \hat{y}_2, \cdots = f_{LM}(Q, \{r_1, r_2, \cdots\}),$$

$$\hat{Y} = f_{LM}(Q, R),$$

where $f_{LM}$ refers to an algorithm with a language model. $\hat{y}_i$ is the binary prediction for each sentence $r_i$ and $\hat{Y}$ is the binary prediction for the entire response $R$, with positive value referring to hallucinated and negative value referring to factual.

## 2.2 KNOWLEDGE HYPERGRAPH

A knowledge hypergraph is used to store and describe the relationships of knowledge statements with a graph structure. Each vertice $v$ refers to an entity. Each hyperedge $e$ connecting any number of vertices refers to a knowledge relating to these entities, which are denoted as $e.nodes$. For example, edge *"The mechanism of Gliclazide is to lower blood glucose by stimulating pancreatic $\beta$-cells to secrete insulin"* connects vertices *"Gliclazide"*, *"blood glucose"*, *"pancreatic $\beta$-cells"*, and *"insulin"* as the statement is directly related to these concepts. We denote a graph as:

$$\mathcal{G} = (\mathbb{V}, \mathbb{E}),$$

where $\mathbb{V}$ and $\mathbb{E}$ respectively refer to the vertice set and the edge set. Compared with vanilla knowledge graphs constructed by triples symbolizing knowledge regarding only two entities, a hyperedge interconnects any number of entities and thus is more suitable for describing complex knowledge (Chen et al., 2024).

## 3 METHODOLOGY

Figure 2 provides an overview of our framework. Given long-form content to be fact-checked, we first extract important entities and statements representing the knowledge to be checked. We then present the framework along with a knowledge hypergraph to iteratively evaluate the factuality of each statement via (1) sampling on the graph to acquire contextual information, (2) evaluating the factuality of each statement and eliciting the intrinsic knowledge by reflection, and (3) updating the graph to retain the elicited thoughts and resolving inconsistencies that might suggest fabrication or induced hallucinations.

## 3.1 STATEMENT EXTRACTION

A common practice to better detect hallucinations is to decompose a long-form text into statements each containing one piece of information (Min et al., 2023; Wei et al., 2024). In our early experiments, we further found that explicitly identifying named entities before the extractions enhances the association of extracted statements to the theme of the given content and alleviates the problem of information missing. Formally,

$$e_1, e_2, \cdots = \text{LM}(\text{Inst}_{ett}, r_1, r_2, \cdots), \tag{1}$$

$$s_1, s_2, \cdots = \text{LM}(\text{Inst}_{state}, e_1, e_2 \cdots, r_1, r_2, \cdots), \tag{2}$$

where $e_i$ is the entity set corresponding to sentence $r_i$. $s_i$ refers to the statements extracted from sentence $r_i$ concerning entities $e_i$. $\text{Inst}_{ett}$ and $\text{Inst}_{state}$ respectively refer to instruction for entity and statement extraction. In practice, the above processes can be achieved in a single chain-of-thought (Wei et al., 2022) reasoning with a prompt with domain expertise.

We then construct the initial knowledge hypergraph as $\mathcal{G}_0 = (\mathbb{V}, \mathbb{E}_0)$, whose vertice set includes all identified entities, i.e. $\mathbb{V} = e_1 \cup e_2 \cup \cdots$, and edge set is empty, i.e. $\mathbb{E}_0 = \varnothing$.

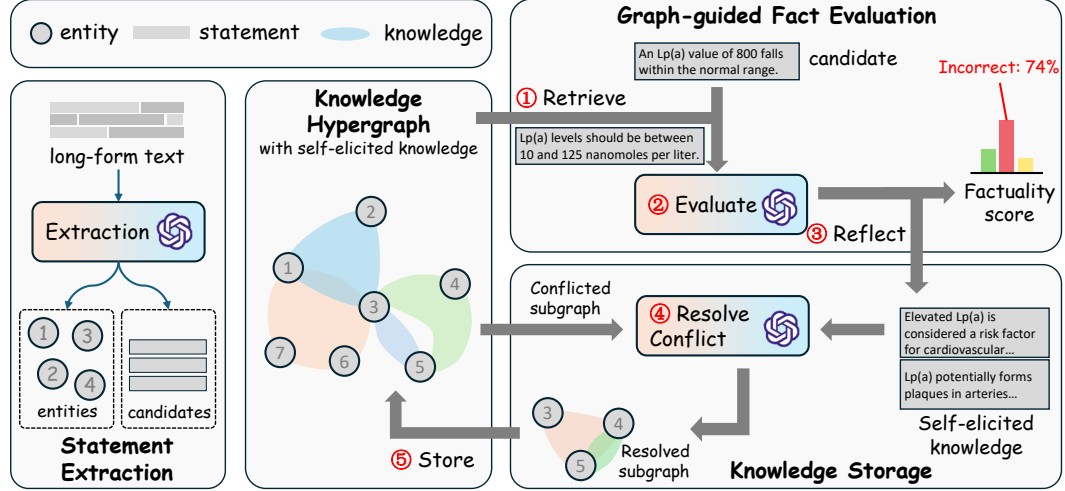

Figure 2: The overall framework of SelfElicit. Given a long-form text, we extract statements and employ an iterative diagram to detect hallucinations via ①sampling relative contextual information and ②evaluating their factuality in order. The intrinsic knowledge of the model is ③elicited by reflecting and then ④adaptively ⑤merged with the existing knowledge hypergraph.

## 3.2 GRAPH-GUIDED SELF-ELICITATION

**Knowledge Sampling**. Given graph $\mathcal{G}_{i-1} = (\mathbb{V}, \mathbb{E}_{i-1})$ retaining self-generated knowledge during prior evaluation of statements $\{s_1, s_2, \cdots, s_{i-1}\}$, a graph sampling procedure is conducted to provide contextual information and intermediate thoughts to the evaluation of current statement $s_i$. Specifically, we extract sub-graphs from $\mathcal{G}_{i-1}$ that are most relevant to $s_i$. A set of relative entities $\mathbb{V}_i$ is first identified by word matching, i.e. $\mathbb{V}_i = \{v_j | v_j \text{ in } s_i, v_j \in \mathbb{V}\}$. Then, sub-graphs with multiple granularities are extracted using the combinations of the relative entities as queries:

$$\hat{\mathbb{V}}_i(k) = \text{Combine}(\mathbb{V}_i, k), \tag{3}$$

$$\hat{\mathbb{E}}_i(k) = \{e | e.nodes == \hat{\mathbb{V}}_i(k), e \in \mathbb{E}_{i-1}\}, \tag{4}$$

$$\hat{\mathbb{E}}_i = \cup\{\hat{\mathbb{E}}_i(k) | \alpha \le k \le \beta\}, \tag{5}$$

where $\text{Combine}(\cdot)$ refers to $k$-length combinations of elements $\mathbb{V}_i$. $\alpha$ and $\beta$ are hyperparameters balancing the relevance and scope. Lower $\alpha$ refers to a more relaxed matching strategy for a wider sampling scope, while higher $\alpha$ refers to a stricter matching strategy for contextual information with stronger relevance. Finally, all sampled edges $\hat{\mathbb{E}}_i$ are linearized to obtain contextual statements $C_i$.

**Fact-Evaluation**. Following (Kadavath et al., 2022; Manakul et al., 2023; Zhao et al., 2024; Tian et al., 2024), we prompt the models to evaluate the correctness of a given statement $s_i$ by asking whether the statement is `True`, `False`, or `Not Sure`. This straightforward prompt has shown relatively stable and competitive performance (Zhao et al., 2024; Mahaut et al., 2024). We prepend the sampled contextual statements $C_i$ to the prompt to leverage semantic continuity for better understanding and reasoning.

The probabilities of `True` and `False` tokens are obtained at the first output token position and normalized. The latter is regarded as the final hallucination score of the statement $s_i$, denoting as $\hat{p}_i$.

**Intrinsic Knowledge Elicitation**. Language models pre-trained on a large corpus abstract the factual knowledge in their weights, i.e. intrinsic knowledge (Petroni et al., 2019). Efforts have been made to elicit the intrinsic knowledge to facilitate fact-checking (Weller et al., 2024; Li et al., 2024b; Manakul et al., 2023; Miao et al., 2024; Mündler et al., 2024; Dhuliawala et al., 2024; Zhao et al., 2024). Nevertheless, we conclude that these methods mostly have complicated reasoning chains, and tend to suffer from induced hallucinations or accumulated inaccuracy, limiting their overall capacity for elicitation. Moreover, we argue that prompting the model to provide reflections on the evaluation is a more convenient method, which guides the model to provide elaborations on its judg-

ment and think of further steps, consequently eliciting the intrinsic knowledge conditioned on the verified statement. The overall fact-evaluation and reflection process is formulated as follows:

$$O_i^{eval}, O_i^{refl} = \text{LM}(C_i, \text{Inst}_{eval}, s_i), \tag{6}$$

where $\text{Inst}_{eval}$ is the evaluation and reflection instruction. In practice, we notice that the reflections $O_i^{refl}$ might include background, coherent thoughts, detailed elaboration, and suggestions relevant to the verified statement. We only keep reflection sentences with objective knowledge with manually crafted rules and LLM prompting, similar to the data preprocessing (Appendix C.3).

### 3.3 Elicited Knowledge Storage

**Graph Updating**. After eliciting intrinsic knowledge conditioned on the statement, we store it in the graph and handle potential knowledge inconsistencies to provide factual contextual information for evaluating subsequent statements. Specifically, the selected reflection $O_i^{refl}$ is first converted into candidate edges by (1) extracting knowledgeable statements from the reflection, (2) identifying entities from $\mathbb{V}$ that verbally matched in each statement as vertices, and (3) creating an edge for each statement. Formally,

$$c_1, c_2, \cdots, c_N = \text{Candidate}(O_i^{refl}), \tag{7}$$
$$\mathbb{V}_j^{new} = \{v | v \text{ in } c_j, v \in \mathbb{V}\}, \quad 1 \le j \le N, \tag{8}$$
$$\mathbb{E}_i^{new} = \{e_j^{new} | e_j^{new}.nodes == \mathbb{V}_j^{new}, 1 \le j \le N\}, \tag{9}$$

where function $\text{Candidate}(\cdot)$ refers to sentence tokenization or model-based knowledge extraction. $e_j^{new}$ refers to a new edge connecting vertices $\mathbb{V}_j^{new}$ and representing one piece of information $c_j$. $N$ is the number of extracted knowledgeable statements. We then iteratively merge each new edge in $\mathbb{E}_i^{new}$ into graph $\mathcal{G}_{i-1}$ to obtain the updated graph $\mathcal{G}_i$:

$$\mathbb{E}_i = \text{Merge}(\mathbb{E}_{i-1}, \mathbb{E}_i^{new}). \tag{10}$$

**Conflicts Resolving**. However, LLMs might produce hallucinations during the reflection process, especially when reflecting on ambiguous or unfamiliar statements. Similar to previous works (Mündler et al., 2024; Yehuda et al., 2024), we notice that in such cases, the generated statements tend to be inconsistent with each other, appearing to have identical entities yet contradictory meanings. The phenomenon of inconsistency can also be found when two sentences from the original response contradict each other, which might indicate faithfulness hallucinations.

To this end, it is crucial to carefully resolve the inconsistencies to avoid the propagation of hallucinations (Zhang et al., 2023a). Specifically, we predict the semantic relationship between the conflictive statements, $e^{new} \in \mathbb{E}_i^{new}$ and $e^{orig} \in \mathbb{E}_{i-1}$, that share identical vertice sets, i.e. $e^{new}.nodes == e^{orig}.nodes$. A Natural Language Inference (NLI) method is utilized to predict their semantic relationship and resolve conflicts:

- **Neutral**: The two statements describe different entities, or different aspects of the same entities, and can coexist. We keep both statements in the updated graph.
- **Entail**: The content of the two statements is identical, describing the same aspect of the same entities, with consistent meaning. We replace the original statement with the new one to avoid duplication.
- **Contradict**. The two statements describe the same aspect of the same entities, but their meanings are directly opposite, presenting a contradiction. In this case, we ask the model to contrast these statements and revise them for a final resolution.

In practice, we can either use a pre-trained NLI model or prompt LLMs to predict the semantic relationships (see Appendix D.3). The resolving process between two conflictive edges is conducted iteratively until all candidate edges in $\mathbb{E}_i^{new}$ have been incorporated into the graph.

To sum up, the knowledge hypergraph is iteratively extended by the elicited knowledge in parallel with the evaluation of statements, by which the semantic continuity information can also be incorporated into the graph, facilitating subsequent evaluation and elicitation.

**Output**. After obtaining the hallucination score $\hat{p}$ for all statements, we aggregate the scores with maximum to obtain the predictions $\hat{y}_i$ for each original sentence $r_i$ and $\hat{Y}$ for the original response $R$. The pseudo-code is shown in Appendix B. The prompts and cases are listed in Appendix F.

Table 1: Full hallucination detection results. S: sentence-level metrics. R: response-level metrics. **Red**: the best. Blue: the second best.

| Methods | Metric | SelfElicit F1 | AUC | IO F1 | AUC | ContextIO F1 | AUC | HistoryIO F1 | AUC | CoT F1 | AUC | CoVE F1 | AUC | FaR F1 | AUC | SelfChkGPT F1 | AUC | ChatProtect F1 | AUC |
|---|---|---|---|---|---|---|---|---|---|---|---|---|---|---|---|---|---|---|---|
| **MedHallu-ZH** | | | | | | | | | | | | | | | | | | | |
| Qwen | S | **0.269** | **0.810** | 0.187 | 0.771 | 0.191 | 0.760 | 0.238 | 0.782 | 0.192 | 0.638 | 0.165 | 0.597 | 0.207 | 0.763 | 0.085 | 0.500 | 0.085 | 0.512 |
| Qwen | R | **0.475** | **0.671** | 0.441 | 0.598 | 0.430 | 0.603 | 0.453 | 0.653 | 0.402 | 0.571 | 0.395 | 0.548 | 0.441 | 0.613 | 0.395 | 0.500 | 0.395 | 0.517 |
| GLM3 | S | **0.228** | **0.798** | 0.182 | 0.756 | 0.153 | 0.733 | 0.213 | 0.781 | 0.131 | 0.564 | 0.170 | 0.661 | 0.139 | 0.702 | 0.085 | 0.494 | 0.134 | 0.611 |
| GLM3 | R | **0.445** | **0.622** | 0.421 | 0.598 | 0.424 | 0.582 | 0.435 | 0.614 | 0.395 | 0.527 | 0.423 | 0.567 | 0.405 | 0.554 | 0.395 | 0.500 | 0.395 | 0.558 |
| **WikiBio** | | | | | | | | | | | | | | | | | | | |
| Qwen | S | - | **0.594** | - | 0.527 | - | 0.587 | - | 0.543 | - | 0.500 | - | 0.527 | - | 0.543 | - | 0.539 | - | 0.512 |
| Qwen | R | - | 0.653 | - | 0.628 | - | 0.522 | - | 0.614 | - | 0.566 | - | 0.524 | - | 0.508 | - | 0.639 | - | **0.657** |
| Llama2 | S | - | 0.556 | - | 0.516 | - | 0.534 | - | 0.477 | - | 0.534 | - | 0.553 | - | 0.506 | - | **0.572** | - | 0.517 |
| Llama2 | R | - | 0.698 | - | 0.559 | - | 0.534 | - | 0.540 | - | 0.531 | - | 0.636 | - | 0.522 | - | **0.708** | - | 0.704 |
| **MedHallu-EN** | | | | | | | | | | | | | | | | | | | |
| Qwen | S | **0.242** | **0.803** | 0.182 | 0.762 | 0.168 | 0.743 | 0.233 | 0.779 | 0.192 | 0.596 | 0.085 | 0.500 | 0.187 | 0.763 | 0.226 | 0.682 | 0.085 | 0.505 |
| Qwen | R | 0.463 | 0.656 | 0.436 | 0.622 | 0.443 | 0.614 | **0.472** | **0.659** | 0.395 | 0.570 | 0.395 | 0.498 | 0.445 | 0.630 | 0.428 | 0.623 | 0.395 | 0.505 |
| Qwen2 | S | **0.282** | **0.820** | 0.275 | 0.805 | 0.247 | 0.802 | 0.254 | 0.811 | 0.211 | 0.636 | 0.259 | 0.672 | 0.217 | 0.784 | 0.232 | 0.675 | 0.087 | 0.523 |
| Qwen2 | R | **0.479** | **0.667** | 0.466 | 0.665 | 0.460 | 0.661 | 0.456 | 0.656 | 0.422 | 0.595 | 0.440 | 0.614 | 0.447 | 0.640 | 0.444 | 0.636 | 0.395 | 0.537 |
| Llama2 | S | **0.181** | **0.748** | 0.137 | 0.697 | 0.139 | 0.705 | 0.133 | 0.667 | 0.142 | 0.594 | 0.085 | 0.499 | 0.140 | 0.709 | 0.103 | 0.561 | 0.136 | 0.550 |
| Llama2 | R | 0.408 | **0.582** | 0.410 | 0.555 | 0.407 | 0.509 | 0.413 | 0.551 | 0.395 | 0.537 | 0.395 | 0.497 | 0.411 | 0.558 | 0.397 | 0.547 | 0.395 | 0.568 |
| Llama3 | S | 0.211 | **0.773** | 0.156 | 0.724 | 0.170 | 0.741 | 0.147 | 0.662 | **0.223** | 0.666 | 0.184 | 0.699 | 0.184 | 0.730 | 0.158 | 0.634 | 0.208 | 0.601 |
| Llama3 | R | 0.447 | 0.622 | 0.406 | 0.546 | 0.405 | 0.572 | 0.413 | 0.605 | **0.449** | **0.626** | 0.421 | 0.562 | 0.422 | 0.586 | 0.417 | 0.613 | 0.414 | 0.600 |
| GPT4o mini | S | **0.329** | 0.682 | 0.185 | 0.560 | 0.183 | 0.564 | 0.250 | 0.597 | 0.279 | 0.686 | 0.277 | **0.703** | 0.085 | 0.520 | 0.135 | 0.623 | 0.085 | 0.512 |
| GPT4o mini | R | **0.494** | **0.668** | 0.395 | 0.559 | 0.395 | 0.574 | 0.395 | 0.586 | 0.487 | 0.661 | 0.488 | 0.658 | 0.395 | 0.521 | 0.395 | 0.603 | 0.395 | 0.505 |
| 1ˢᵗ count | S | **13** | | 0 | | 0 | | 0 | | 1 | | 1 | | 0 | | 1 | | 0 | |
| 1ˢᵗ count | R | **9** | | 0 | | 0 | | 3 | | 2 | | 0 | | 0 | | 1 | | 1 | |

## 4 EXPERIMENTS

### 4.1 EXPERIMENTAL SETUPS

We conduct long-form hallucination detection experiments on two medical datasets (MedHallu-ZH and MedHallu-EN dataset, see Appendix C.2), and a biography dataset (WikiBio (Manakul et al., 2023)) with the following off-the-shelf language models: Qwen1.5-7B-chat (`Qwen` (Bai et al., 2023)), Qwen2.5-7B-Instruct (`Qwen2` (Bai et al., 2023)), ChatGLM3-6B (`GLM` (GLM et al., 2024)), Llama2-7B-chat (`Llama2` (Touvron et al., 2023)), Llama-3-8B-Instruct (`Llama3` (AI@Meta, 2024)), and GPT4o-mini. All language models use greedy decoding (temperature=0) during text generation for stable outputs. All experiments are conducted with transformers (Wolf et al., 2020) 4.43.0 on a Centos machine with Nvidia A800-80G GPUs.

**Baselines**. We compare our method with the following baselines, including classic self-eval (IO (Kadavath et al., 2022; Mahaut et al., 2024)), long-form enhanced methods (ContextIO, HistoryIO), and methods with various elicitation approaches: chain-of-thought (CoT (Wei et al., 2022) and FaR (Zhao et al., 2024)), self-ask (CoVE (Dhuliawala et al., 2024)), and self-consistency (Self-CheckGPT (Manakul et al., 2023), ChatProtect (Mündler et al., 2024)). For all methods, we use an identical IO prompt after their original procedures to obtain the hallucination score for a fair comparison (i.e. only elicitation approaches are different). The details are listed in Appendix C.1.

**Metrics**. Hallucination detection is a classification task, where positive labels refer to non-factual statements. We use *F1* and *AUROC* as metrics, for sentence-wise and response-wise predictions. Since the threshold variance affects the metrics (Huang et al., 2024), we search for the best threshold values with the highest sentence/response F1 metrics independently on the validation set and regard non-factual scores larger than the thresholds as positive predictions on the test set.

### 4.2 MAIN RESULTS

Table 1 shows the overall detection results. Methods requiring multi-step reasoning (CoT, CoVE, SelfCheckGPT, and ChatProtect) generally have inferior performance compared with other methods. This observation is partial because the primary benefit of multi-step reasoning comes in the ability to execute symbolic steps and track the output (Sprague et al., 2024), rather than directly assessing the factuality, leading to limited performance gain. Moreover, we observe that the inaccuracies and hallucinations (e.g. information missing when generating questions for CoVE and triple ambiguity for ChatProtect) accumulate as the reasoning steps increase, resulting in their overall limited capacity to fully utilize the models' intrinsic knowledge. On the contrary, IO and ConfScore

use more straightforward prompts to utilize the models' knowledge, consequently reducing the risk of inaccuracy accumulation and resulting in their better overall performance.

Moreover, context-argument methods (ContextIO and HistoryIO) show better performance than vanilla IO, which proves that using in-context information can benefit the understanding of the current statement. The better performance of HistoryIO suggests that the generated reflections on prior statements might already include some intrinsic knowledge expressed verbally, which reduces the reasoning burden of the current evaluation.

It can also be observed that the latest-generation models (Qwen2 and Llama3) outperform their previous-generation counterparts (Qwen and Llama), which is owed to their stronger capability. With appropriate algorithms, the performance of previous-generation models surpasses the latest-generation models (e.g. SelfElicit+Llama2 > IO+Llama3), highlighting the importance of effective knowledge elicitation methods and hallucination detection frameworks.

These observations motivate us to carefully guide the model to express its intrinsic knowledge and to avoid hallucination accumulation. Conceptually, our method uses self-generated thoughts from prior statements to promote the elicitation and resolve the semantic inconsistencies between the elicited and the existing knowledge, which helps to avoid the snowballing of hallucinated content. Moreover, the iteratively updated diagram also benefits the model in statement understanding and evaluation by capturing semantic continuity. The synergy of these components eventually results in the superior performance of SelfElicit. Statistically, our method outperforms the best baselines over 5.1%/9.4% on MedHallu-zh with Qwen, 5.6%/4.0% on MedHallu-zh with ChatGLM, 5.3%/4.1% on MedHallu-en with Qwen, and 1.0%/-0.6% on MedHallu-en with Llama on sentence/response-wise AUC metrics. These results demonstrate the overall effectiveness of our method.

## 4.3 Elicitation Quality

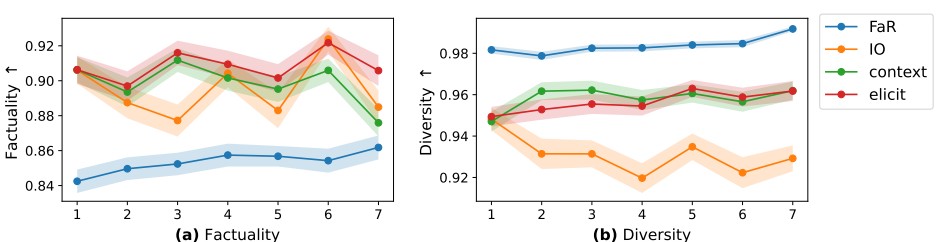

Figure 3: Factuality and diversity of elicited knowledge with different elicitation methods. x-axis refers to the statements' serial number. Higher metrics are better.

We take a deeper look into the elicitation by comparing the factuality and diversity of the elicited knowledge with different elicitating methods: (1) *FaR*: generating relevant information before the evaluation, (2) *IO*: generating reflections without context, (3) *context*: generating reflections with prior statements as context, and (4) *elicit*: generating reflections with previously generated thoughts as context. The reflections are generated with Qwen on the MedHallu-zh dataset and assessed with GPT4 on their factuality (whether the reflection is factual) and diversity (whether the reflection is different from the statement). The results averaged based on the statements' serial numbers are shown in Figure 3 with variance. Results with larger sequence numbers are discarded due to the limited number of statements.

We observe that (1) the information generated with *FaR* has the lowest factuality rates and highest diversity, indicating that it has a higher risk of fabricating its intrinsic knowledge, which explains the occasional performance degradation of FaR compared to IO in Table 1. We owe that using calibration-based evaluation before expressing related knowledge, rather than reversing their order, will prompt the model to reason over the correctness of the statement, which restricts the diversity but alleviates fabrications. (2) Moreover, prefixing context before reflection (*context*) improves both the factuality and diversity of the generated statements in most cases (v.s *IO*), which shows that leveraging the semantic continuity can facilitate the model to understand the statements and provide more faithful and comprehensive reasoning over the knowledge. (3) *Elicit* shows a similar trend with *context* and consistently outperforms. The observation indicates that the self-elicitation mechanism has a similar ability to capture continuity and might suggest that self-generated thoughts

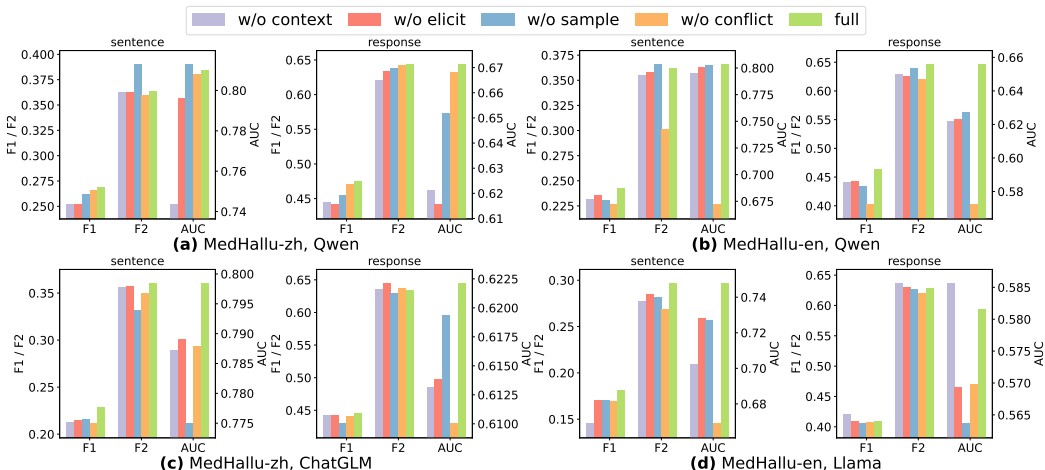

Figure 4: Full ablated results. Higher metrics are better.

are more favorable than the given statements to catalyze further reasoning over related knowledge. In summary, these results have demonstrated that self-elicitation effectively improves the expression of intrinsic knowledge with better faithfulness and comprehensiveness, which provides insights into why self-elicitation works.

## 4.4 ABLATION STUDY

We conduct an ablation study to demonstrate the effectiveness of each component of our method. The variants include: (1) *w/o context*: evaluating the statements without sampled contextual knowledge, which is equivalent to vanilla self-eval (Kadavath et al., 2022; Mahaut et al., 2024), (2) *w/o elicit*: using prior statements rather than reflections as context, (3) *w/o sample*: linearizing the entire graph rather than sampling relevant knowledge as context, (4) *w/o conflict*: merging all new edges without inconsistencies mitigation, and (5) *full*: full SelfElicit method with all components.

Figure 4 shows the results of all variants. We have the following observations. (1) *w/o context* shows a salient performance degradation compared with other variants in a majority of cases, which provides an intuitive demonstration of the contextual understanding in long-form content. However, in several cases (e.g. response-wise metrics in sub-figure (d)), variants with context provide inferior performance. We have conducted a manual review of the predictions and found that the model sometimes misunderstood the task of fact-checking, which we believe can be largely solved with stronger models (e.g. Qwen in sub-figure (b)). (2) Variants without sampling or conflict mitigation provide relatively inferior performance compared to *full*, even perform worse in several cases (e.g. AUC in sub-figure (b,c,d)). We also observe that providing irrelevant or self-contradictory context to the models will greatly disturb their focus and affect their reasoning, demonstrating the importance of knowledge sampling and conflict mitigation components. (3) Versus all ablated variants, the full method generally provides the best performance. The full method outperforms 4.0% over *w/o context*, 3.0% over *w/o elicit*, 2.1% over *w/o sample*, and 6.5% over *w/o conflict* on average, highlighting the synergistic effect of integrating all the constituted parts.

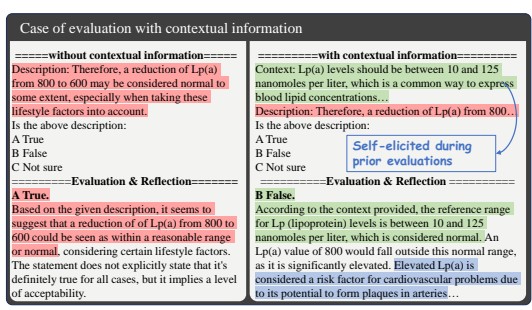

Figure 5: A comparison of evaluation and reflection without(left) or with(right) contextual information. Red : non-factual content. Green : factual content. Blue : newly elicited content.

## 4.5 CASE STUDY

We show a case of evaluation with/without contextual information in Figure 5. Figure 5 left shows that the model has difficulty evaluating the statement since the reasoning includes eliciting intrinsic

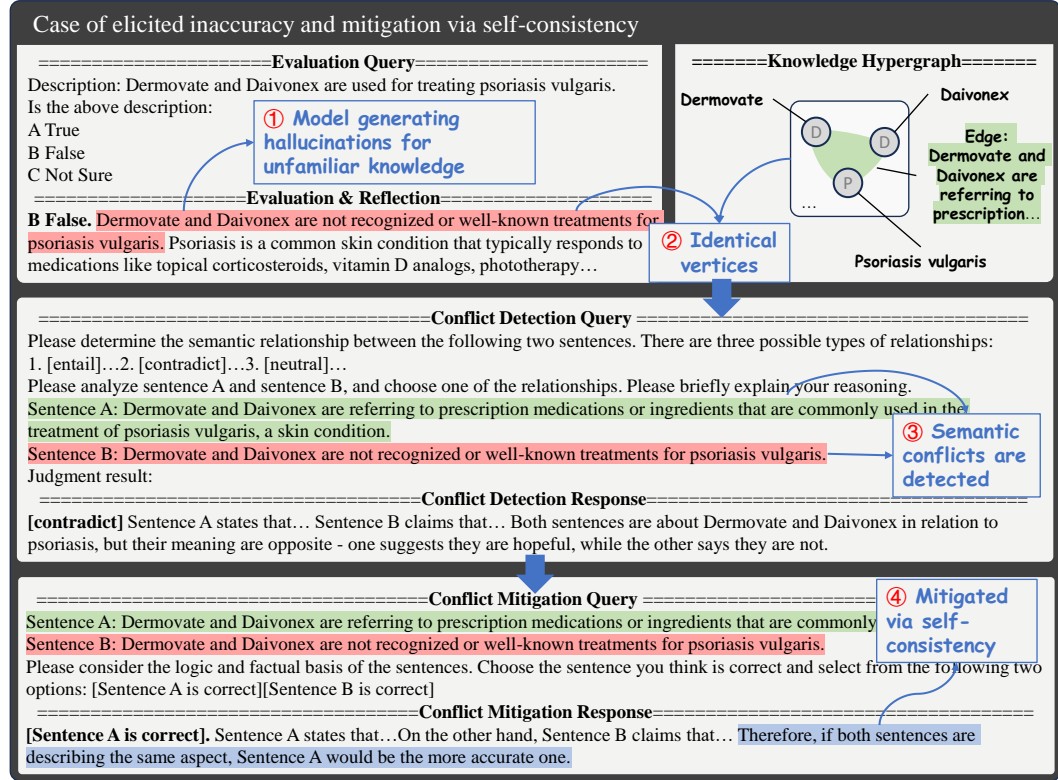

Figure 6: A showcase of generating inaccurate reflection and how the accumulation of inaccuracy is mitigated by conflict detection. Red : non-factual content. Green : factual content.

knowledge about the normal range of Lp(a) and then comparing the values. Figure 5 right shows that self-elicited thoughts during the evaluation of prior statements provide direct information (the normal range of Lp(a)) to facilitate the evaluation of the current statement. A piece of intrinsic knowledge about the indication of Lp(a)( blue ) is also elicited as long as the evaluation.

Figure 6 shows another case of the model failing to evaluate the factuality due to unfamiliarity with specific knowledge and how the accumulation of inaccuracy is mitigated by conflict detection. Since Dermovate and Daivonex are trade names that are less exposed than their pharmaceutical names, the model ①fails to fact-check the statement and generates erroneous reflections ( red ). Such generated hallucinations will accumulate and finally affect the reasoning of subsequent evaluations. However, since the erroneous reflection ②shares an identical vertice set (Dermovate, Daivonex, and Psoriasis) with existing edges in the hypergraph, the ③NLI-based component is activated and predicts their semantic contradict. Finally, the conflict is ④mitigated to avoid the accumulation of errors via LLM reasoning.

## 5 DISCUSSIONS

### 5.1 INFERENCE COSTS

Table 2 and Appendix E.2 show the inference costs for all methods. We observe that SelfCheckGPT, ChatProtect, and CoVE have lower efficiency in both the number of model calls and tokens generated since these methods require multiple reasoning steps to elicit the knowledge. We also observe that SelfElicit has a cost that ranks moderately among all methods while achieving on average the best performance. Conceptually, the sampling and the merging of new edges (①⑤ in Figure 2) are both rule-based and the reflection and resolving procedure (③④ in Figure 2) contribute the majority of overhead. These results demonstrate that it might be unnecessary to design complicated reasoning steps to prompt the expression of intrinsic knowledge and using self-elicitation can have better performance and efficiency during hallucination detection.

Table 2: Inference costs for all methods with the Qwen model. `Perform.`: average AUC metrics. `#Call`: number of LLM calls. `#Token`: number of generated tokens.

| Dataset | Method | Relative Perform.↑ | #Call↓ | Relative #Call↓ | #Token↓ (k) | Relative #Token↓ |
|---|---|---|---|---|---|---|
| MedHallu-zh | IO | -7.9% | 7,552 | -39.4% | 390 | -61.7% |
| | ContextIO | -8.1% | 7,552 | -39.4% | 399 | -60.9% |
| | HistoryIO | -3.1% | 7,552 | -39.4% | 370 | -63.7% |
| | CoT | -18.4% | 7,552 | -41.6% | 734 | -28.1% |
| | CoVE | -22.7% | 36,852 | +196.0% | 1,828 | +79.2% |
| | FaR | -7.1% | 14,104 | +13.3% | 2,309 | +126.4% |
| | SelfCheckGPT | -32.5% | 130,912 | +951.3% | 13,711 | +1244.0% |
| | ChatProtect | -30.5% | 138,758 | +1014.3% | 5,703 | +459.0% |
| | SelfElicit | - | 12,452 | - | 1,020 | - |
| MedHallu-en | IO | -5.2% | 7,422 | -37.3% | 636 | -54.7% |
| | ContextIO | -7.0% | 7,422 | -37.3% | 657 | -53.2% |
| | HistoryIO | -1.2% | 7,422 | -37.3% | 489 | -65.2% |
| | CoT | -20.1% | 7,422 | -37.3% | 1,096 | -22.0% |
| | CoVE | -31.6% | 38,696 | +226.7% | 2,484 | +76.8% |
| | FaR | -5.9% | 14,104 | +19.1% | 2,752 | +95.9% |
| | SelfCheckGPT | -10.5% | 131,066 | +1006.5% | 10,828 | +670.9% |
| | ChatProtect | -30.8% | 164,010 | +1284.6% | 6,398 | +355.5% |
| | SelfElicit | - | 11,845 | - | 1,405 | - |

## 5.2 CONNECTION WITH RAG

SelfElicit and retrieval-argument generation (RAG) (Jin et al., 2024; Luo et al., 2024; Sun et al., 2024) share some similarities in their schemas: sampling relative knowledge from a knowledge graph to facilitate the down-streaming tasks. Recent works (Sansford et al., 2024; Yuan et al., 2024; Niu et al., 2024) have demonstrated the performance gain to incorporate external knowledge graphs for hallucination detection. Differently, our work organizes a knowledge graph elicited from the model itself, rather than relying on external databases. Moreover, compared with RAG methods where databases are stand-alone, the self-elicited knowledge hypergraph in our framework is dependent on the model and evolves in parallel with the evaluation process. Theoretically, our method is orthogonal to these RAG methods and can be integrated with these methods into a unified design, which might further benefit both the elicitation and hallucination detection.

## 5.3 LIMITATIONS

Some of the limitations are: (1) this paper primarily focuses on technological methods to elicit the intrinsic knowledge of models, leaving the question of whether LLMs either abstract knowledge over linguistic forms or merely memorize statements (Carlini et al., 2022) to future works. (2) Due to the lack of large-scale, long-form datasets with domain expertise, we focus on two medical datasets collected from an online QA platform and a biography dataset. Although the models are not specialized for specific domains, conducting experiments on other domains would provide more comprehensive and credible conclusions. (3) Since the capacity of the models theoretically restricts the performance upper bound, methods for continual improvements remain an open question. (4) The sampling and conflict detection strategies will fail in some specific cases and such failure might accumulate during the iteration.

## 6 CONCLUSION

In this paper, we have investigated the task of detecting hallucinations from long-form content. Existing methods predominantly fall short of comprehensively eliciting the intrinsic knowledge of models and overlook the semantic continuity within long-form content. To address these issues, we present a novel framework, SelfElicit, that uses self-generated thoughts from prior statements to elicit the models' intrinsic knowledge. It is integrated with a knowledge hypergraph to enable effective knowledge organization via retention, deduplication, and inconsistency mitigation, therefore synergizing self-elicitation and contextual understanding in a unified diagram. Experiments on real-world, multilingual datasets with modern large language models have shown the effectiveness of self-elicitating and demonstrated the superiority of the proposed framework.

ETHICAL CONSIDERATIONS

The statements and examples provided in this paper are intended for demonstration purposes only and may contain non-factual information. Our intent is to illustrate concepts rather than present verified facts. Readers are strongly advised to consult with professional healthcare providers or academic experts before taking any medical actions.

REPRODUCABILITY STATEMENT

We provide the source code of the implementations of all methods in `https://anonymous.4open.science/r/SelfElicit-DFCE`. Due to privacy concerns, the datasets used in our experiments can not be included during the peer review process. However, we are committed to making the data publicly available upon the acceptance of our paper. The large language models used in our work are publicly accessible online: Qwen1.5-7B-chat[2], Qwen2.5-7B-Instruct[3], ChatGLM3-6B[4], Llama2-7B-Chat[5], and Llama3.1-8B-Instruct[6].

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

# A RELATED WORKS

## A.1 HALLUCINATION DETECTION

**Retrieval-argument methods**. Extracting relevant knowledge from external authentic database and incorporating it with the query is a common way of detecting hallucination (Min et al., 2023; Tian et al., 2024; Gou et al., 2024; Li et al., 2024b; Xia et al., 2024). (Li et al., 2023b; Yu et al., 2023; Wei et al., 2024) proposed to update the retrieval results with LLM until the retrieved documents adequately support answering the questions. (Kamoi et al., 2024), (Yuan et al., 2024) and (Sansford et al., 2024) extracted keywords as entities and knowledge as triples and retrieved reference triples from knowledge graphs or texts. Additionally, (Yue et al., 2024) contrasted the supportive arguments and refuting arguments derived from retrieval evidence.

**Innerstate-based methods**. Innerstate-based methods aim to understand the hallucination within the hidden activations of deeper model layers (Azaria & Mitchell, 2023; Zhang et al., 2024b; Wang et al., 2024). They usually required probes pre-trained on a specific dataset to detect the hallucinations (Li et al., 2023a; Zhang et al., 2024a).

**Uncertainty-based methods**. We categorize existing uncertainty-based hallucination detection methods into three categories. (1) Some methods focus on the token probabilities of white-box LLMs. (Kadavath et al., 2022; Tian et al., 2024) proposed a calibration-based method to evaluate the correctness of the content with multiple-choice questions. Extending the token entropy estimation (Manakul et al., 2023) with keyword focusing, (Zhang et al., 2023b) proposed to penalize the attention score of the hallucinated token to avoid snowballing (Zhang et al., 2023a). FaR (Zhao et al., 2024) elicited the intrinsic knowledge relevant to the query and reflected on the knowledge to improve the calibration. (2) Some methods propose to ask LLMs to express their uncertainty verbally (Mahaut et al., 2024). (Tao et al., 2024) leverages reinforcement learning guided by a tailored dual-component reward function. (3) Other methods aim at the semantic consistency over sentences (Kuhn et al., 2023; Manakul et al., 2023; Mündler et al., 2024; Miao et al., 2024). Self-CheckGPT (Manakul et al., 2023) and (Kuhn et al., 2023) and (Farquhar et al., 2024) estimated the variance of the meaning of generated content. (Cohen et al., 2023) discovered the inconsistencies with the interaction between LLMs. InterrogateLLM (Yehuda et al., 2024) reversed the query-response pair and estimated the variation of reconstructed queries for semantic uncertainty. ChatProtect (Mündler et al., 2024) and SelfCheck (Miao et al., 2024) detected hallucinations by comparing the original content and the regenerated one. EVER (Kang et al., 2023), CoVE (Dhuliawala et al., 2024), (Zhang et al., 2024c), (Farquhar et al., 2024), and QuestGen (Setty & Setty, 2024) generated questions corresponding to each fact within the content, answered the generated question, and measured the coherence between the answer and the original content.

Compared with the above works, our method uses self-generated thoughts as a catalyst to elicit intrinsic knowledge, without external databases, finetuning, or complex multi-step reasoning, while the iterative schema can capture the semantic continuity of long-form content.

## A.2 LARGE LANGUAGE MODELS WITH KNOWLEDGE GRAPHS

Efforts have been made to facilitate large language models for reasoning or factuality with knowledge graphs. GoT (Besta et al., 2024) used a graph structure to guide the reasoning of LLMs. (Yuan et al., 2024) proposed to extract knowledge graphs from external text databases and regarded fact-checking as a task of NLI. GraphRAG (Edge et al., 2024) built a graph-based text index by deriving entity knowledge graphs from the source documents and generating summaries for hierarchical graph communities. RoG (Luo et al., 2024) synergized LLMs reasoning with KGs to improve the ability of knowledge traceability and knowledge correctability. ToG (Sun et al., 2024) and Graph-CoT (Jin et al., 2024) treated the LLM as an agent to interactively explore related entities and relations on KGs and perform reasoning based on the retrieved knowledge. Re-KGR (Kamoi et al., 2024) and StructGPT (Jiang et al., 2023) leveraged knowledge graphs as external databases and directly retrieved reference information for factual QA. (Sansford et al., 2024) converted the response into a candidate knowledge graph and fact-checked each individual triple in the graph. Compared with the above methods, our method does not rely on external knowledge graphs but uses self-elicited knowledge to construct the graph to facilitate hallucination detection.

## B ALGORITHM

Algorithm 1 shows the pseudo-code of SelfElicit.

---

**Algorithm 1:** Self-elicitation Procedure.

---

**Input** : Sentences $\{r_1, r_2, \cdots\}$, a language model LM, a NLI Model NLI.
**Output:** Sentence-wise non-factual scores $\hat{y}_1, \hat{y}_2, \cdots$, and response-wise score $\hat{Y}$.

    /* Extract entities and statements             */
1  $s_1, s_2, \cdots, e_1, e_2, \cdots \leftarrow \text{LM}(r_1, r_2, \cdots)$;
    /* Graph-guided self-elicitation           */
2  Initialize graph $\mathcal{G}_0$ with vertice set $\mathbb{V} \leftarrow e_1 \cup e_2 \cup \cdots$, and edge set $\mathbb{E}_0 \leftarrow \varnothing$;
3  **for** $s_i \in \{s_1, s_2, \cdots\}$ **do**
      /* Knowledge sampling             */
4     **for** $k \in [\alpha, \beta]$ **do**
5         | Sample $\hat{\mathbb{E}}_i(k)$ from graph $\mathcal{G}_{i-1}$ with related vertives $\hat{\mathbb{V}}_i(k)$;
6     **end**
7     Aggregate all $\hat{\mathbb{E}}_i(k)$ and linearize to context $C_i$;
      /* Fact-evaluation & Elicitation           */
8     Evaluate $s_i$ given context $C_i$ with LM, obtaining score $\hat{p}_i$ and reflection $O_i^{refl}$;
      /* Graph Update             */
9     Obtain new edges $\mathbb{E}_i^{new}$ from reflection $O_i^{refl}$;
10    $\mathbb{E}^{orig} \leftarrow \mathbb{E}_{i-1}$;
11    **for** $e \in \mathbb{E}_i^{new}$ **do**
12       $\mathbb{E}^{temp} \leftarrow \varnothing$;
13       **if** *e has identical vertice set to any edge* $\bar{e} \in \mathbb{E}^{orig}$ **then**
14          $rel \leftarrow \text{NLI}(e, \bar{e})$;
15          **if** *rel is 'entail'* **then** Add $e$ to $\mathbb{E}^{temp}$ ;
16          **else if** *rel is 'neutral'* **then** Add $e$ and $\bar{e}$ to $\mathbb{E}^{temp}$ ;
17          **else**             /* mitigate conflicts */
18            $\hat{e} \leftarrow \text{LM}(e, \bar{e})$;
19            Add $\hat{e}$ to $\mathbb{E}^{temp}$;
20       **else**
21          Add $e$ to $\mathbb{E}^{temp}$;
22       **end**
23       $\mathbb{E}^{orig} \leftarrow \mathbb{E}^{temp}$
24    **end**
25    Update graph $\mathcal{G}_i$ with edge set $\mathbb{E}^{orig}$;
26 **end**
27 Obtain sentence predictions $\hat{y}$ by aggregating scores from statements;
28 Obtain response prediction $\hat{Y}$ by aggregating scores from sentences;

---

## C EXPERIMENTAL DETAILS

### C.1 BASELINES

Our comparison includes representative methods that focus on retrieval-free, training-free methods for post-generation fact-checking, including classic self-eval,

- **IO**(Kadavath et al., 2022): Probability of `False` token following a query whether the statement is factual or not.

long-form argument methods,

- **ContextIO**: Prior evaluated statements are prefixed as contextual information.

- **HistoryIO**: Historical information (queries and responses) of prior evaluations are prefixed as contextual information.

and methods with various elicitation approaches (chain-of-thought (Wei et al., 2022), self-ask, and self-consistency).

- **CoT**(Wei et al., 2022): Prompting to evaluate the factuality of the given statement after step-by-step reasoning.

- **CoVE**(Dhuliawala et al., 2024): Generating verification questions given the statement, answering the questions independently, and summarizing for final evaluation.

- **FaR**(Zhao et al., 2024): Eliciting the knowledge relevant to the statement from models and asking models to reflect on them to generate the final answer.

- **SelfCheckGPT**(Manakul et al., 2023): Querying to assess whether the statement is supported by stochastic context answering the original user query.

- **ChatProtect**(Mündler et al., 2024): Extracting knowledge triples, cloze triples, and predicting the contradiction between the given and the new statements.

We have excluded some related methods designed to quantify the uncertainty of generator LM during generating statements rather than analyzer LM on the post-generation stage (Fadeeva et al., 2024; Zhang et al., 2023b; Yehuda et al., 2024) and methods required training on specific datasets before detecting hallucinations (Zhang et al., 2024a; Wang et al., 2024; Li et al., 2023a; Chuang et al., 2024) or focusing on retrieval-argument generation (Min et al., 2023; Tian et al., 2024; Li et al., 2024b; Xia et al., 2024). For all methods, we use an identical IO prompt after their original procedures to obtain the hallucination score for a fair comparison, i.e. only elicitation approaches are different.

## C.2 DATASET

We have collected a substantial dataset, namely `MedHallu`, by collecting genuine user queries and the corresponding responses generated by LLMs from an online healthcare QA platform. This corpus mainly encompasses chronic diseases, cancer, and psoriasis and includes a Chinese version (with postfix `zh`) and an English version (with postfix `en`). The query-response pairs are preprocessed with the following steps to obtain response-wise and sentence-wise hallucination labels.

**Step 1: Parsing**. The long-form response is first segmented into sentences by punctuation. Then, following (Wei et al., 2024), GPT-4 is used to split sentences into atomic claims, which refers to a fundamental unit for a piece of information.

**Step 2: Labeling**. We ask medical experts to label whether each LLM-generated response includes any factual error or misunderstands the user query. Then, GPT-4 is used to label each sentence given the response-wise human labels to obtain sentence-wise labels and claim-wise labels. The labeling prompt is shown in Figure 13. We carefully check every positive sentence/claim labeled by GPT-4 such that they actually include factual errors.

**Step 3: Multi-language**. We use GPT4 to translate the original Chinese QA pairs, sentences, and claims into English. The prompt is constituted of the original user query and the LLM-generated response and includes instructions to ask the LLM to take special care of the medical terminologies. The translation prompt is shown in Figure 12.

**Step 4: Remove Duplicate & Split**. Samples with duplicated queries and responses are removed. Then all samples are randomly shuffled and split into train/validation/test sets with a ratio of 0.6/0.1/0.3.

Table 3 shows the statistics of the datasets. Table 4 show a sample from `MedHallu-en`. For the WikiBio[7] dataset, the split ratio is 0/0.3/0.7 and both minor and major inaccuracies are regarded as non-factual.

---

[7] https://huggingface.co/datasets/potsawee/wiki_bio_gpt3_hallucination

Table 3: Statistics of `MedHallu-zh` and `MedHallu-en` datasets. `pos%`: proportion of non-factuality. `#Sentence/Sample`: number of sentences per sample. `#Claim/Sample`: number of claims per sample.

| Split | Sample | Total (pos%) Sentence | Claim | #Sentence/Sample avg. | min. | max. | #Claim/Sample avg. | min. | max. |
|---|---|---|---|---|---|---|---|---|---|
| Train | 1,622 (27.6%) | 10,688 (5.3%) | 11,677 (4.7%) | 6.59 | 1 | 22 | 7.20 | 1 | 30 |
| Validate | 270 (24.1%) | 1,809 (4.4%) | 1,970 (3.8%) | 6.70 | 1 | 21 | 7.30 | 2 | 23 |
| Test | 812 (24.6%) | 5,534 (4.4%) | 5,984 (4.1%) | 6.81 | 1 | 30 | 7.37 | 1 | 26 |

**User:**
When should Gliclazide be taken? Why?
**Response:**
Gliclazide is an oral hypoglycemic medication. It is suitable for adult type 2 diabetes patients whose blood sugar cannot be controlled by diet alone. Generally, Gliclazide can be taken at any time of the day, regardless of whether it is on an empty stomach or after meals. This is mainly because the mechanism of action of Gliclazide is to lower blood sugar by stimulating the secretion of insulin from pancreatic $\beta$-cells, and this process is not affected by meal intake. It should be noted that the specific timing and dosage of medication should be determined based on individual differences and the doctor's advice. In addition, during the use of Gliclazide, it is necessary to regularly monitor blood sugar levels and adjust the dosage according to medical advice. Do not increase or decrease the dosage or stop taking the medication on your own to avoid adverse reactions such as hypoglycemia.
**Labeling:** ✗ Incorrect. Gliclazide should be taken before each meal.

**Sentences:**
1. Gliclazide is an oral hypoglycemic medication✓
2. It is suitable for adult type 2 diabetes patients whose blood sugar cannot be controlled by diet alone✓
3. Generally, Gliclazide can be taken at any time of the day, regardless of whether it is on an empty stomach or after meals✗
4. This is mainly because the mechanism of action of Gliclazide is to lower blood sugar by stimulating the secretion of insulin from pancreatic $\beta$-cells, and this process is not affected by meal intake✗
5. It should be noted that the specific timing and dosage of medication should be determined based on individual differences and the doctor's advice✓
6. In addition, during the use of Gliclazide, it is necessary to regularly monitor blood sugar levels and adjust the dosage according to medical advice✓
7. Do not increase or decrease the dosage or stop taking the medication on your own to avoid adverse reactions such as hypoglycemia✓
**Claims:**
1. Gliclazide is an oral hypoglycemic medication✓
2. Gliclazide is suitable for adult type 2 diabetes patients whose blood sugar cannot be adequately controlled by diet alone✓
3. Gliclazide can be taken at any time of the day✗
4. Gliclazide can be taken either on an empty stomach or after meals✗
5. The mechanism of action of Gliclazide is to lower blood glucose by stimulating pancreatic $\beta$-cells to secrete insulin✓
6. The action process of Glargine is not affected by food intake✓
7. The specific timing and dosage of Gliclazide medication should be determined based on individual differences and the doctor's recommendations✓
8. During the use of Gliclazide, it is necessary to regularly monitor blood sugar levels and adjust the dosage according to the doctor's instructions✓
9. Do not adjust the dosage or discontinue the medication on your own when using Gliclazide✓
10. Adjusting the dosage of Gliclazide on your own may lead to adverse reactions such as hypoglycemia✓

Table 4: A sample of the dataset in MedHallu-en. `Labeling` refers to the human annotation of the response.

### C.3 Data Preprocessing

As stated in previous works (Deng et al., 2024), the sentences might include information irrelevant to the central idea of the document. Verifying all information is inefficient and even misleading since some statements are simple repetitions of the user query or include subjective thoughts that are not directly relevant to the concept of factuality. To this end, we identify sentences that contain check-worthy statements, including assertions and thoughts regarding objective knowledge. Specifically, we provide the LLMs instructions and few-shot samples with domain-specific expertise and ask them to judge whether a sentence includes any objective knowledge. The selected check-worthy sentences are denoted as $\{r_1, r_2, \cdots\}$. The detailed prompt can be found in the Appendix F.

## D    More Experiments

### D.1    Model Scalability

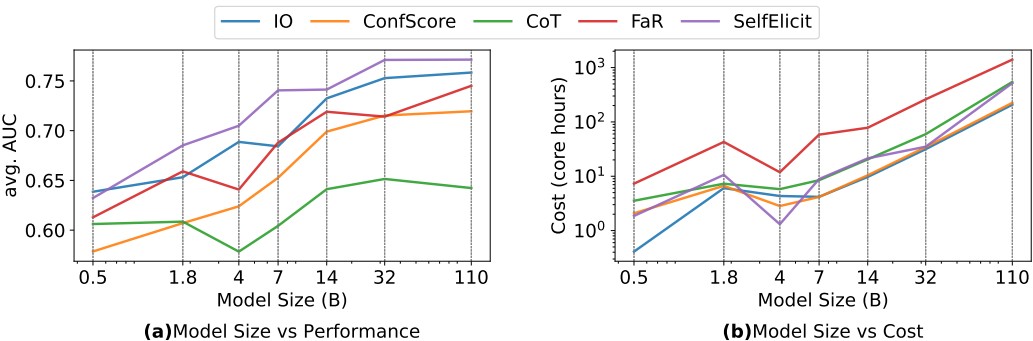

**(a)** Model Size vs Performance          **(b)** Model Size vs Cost

Figure 7: Performance and cost with different model scales. We use logarithmic coordinates for both x-axes and y-axis in sub-figure (b).

We study the relationship between model scale and performance. We choose methods with preferable performance and efficiency (IO, ConfScore, CoT, FaR, and SelfElicit) for comparison and use the Qwen1.5-chat (Bai et al., 2023) family with model sizes 0.5, 1.8, 4, 7, 14, 32, and 110.

The scaling of the performance and cost is shown in Figure 7. We have the following observations. (1) The trend generally follows the scaling law that larger models tend to have better performance and the inference costs also increase nearly linearly with the model size. However, the performance seems to be saturated for the 110B models, having an insignificant increase. (2) We notice a salient performance and cost degradation of the 4B model and a slightly higher cost for the 1.8B model. After manually checking the output, we found that the average output length of the 1.8B models is much longer than that of the 4B model. We owe it to the models' preference obtained during pre-training, rather than during hallucination detection. (3) Both the 7B and 14B models achieve a good balance between performance and cost. Therefore, we choose the 7B model or models with a similar scale to conduct all experiments in this paper. (4) Comparing all baselines, our SelfElicit almost achieves the best performance with all model scales, while having relatively similar inference with CoT.

### D.2    Hyperparameter Sensitivity

By changing the $\alpha$ and $\beta$ hyper-parameters in Equation 3, we can change the sampling scope from the knowledge hypergraph. We conduct experiments to investigate the choices of these hyper-parameters, and matching strategy. Matching strategy `strict` refers to sample an edge iff the query $\mathcal{V}_i(k)$ exactly match the vertice set of an edge, i.e. $e.nodes == \mathcal{V}_i(k)$. `relax` refers to sample an edge if the query $\mathcal{V}_i(k)$ is a subset of the vertices of an edge, i.e. $e.nodes \in \mathcal{V}_i(k)$, providing a wider sampling scope.

Table 5 shows the result with different $\alpha$-$\beta$ pairs. We set the maximum value of both hyperparameters to 3 practically, since we found that combinations of more than 3 entities rarely sample any

Table 5: Results with different $\alpha$-$\beta$ pairs with Qwen on MedHallu-zh.

| Match | $\alpha$ | $\beta$ | sentence | | | paragraph | | |
|---|---|---|---|---|---|---|---|---|
| | | | F1 | F2 | AUC | F1 | F2 | AUC |
| strict | 1 | 1 | **0.272** | **0.373** | 0.794 | 0.458 | 0.642 | 0.656 |
| | 1 | 2 | 0.265 | 0.371 | **0.815** | 0.452 | 0.639 | 0.651 |
| | 1 | 3 | 0.269 | 0.364 | 0.810 | **0.475** | **0.643** | **0.671** |
| | 3 | 3 | 0.242 | 0.347 | 0.783 | 0.434 | 0.621 | 0.611 |
| relax | 1 | 1 | 0.237 | 0.355 | 0.735 | **0.461** | 0.638 | 0.635 |
| | 1 | 2 | **0.264** | **0.373** | **0.816** | 0.453 | **0.639** | **0.655** |
| | 1 | 3 | **0.264** | 0.368 | 0.814 | 0.452 | **0.639** | 0.651 |
| | 3 | 3 | 0.255 | 0.353 | 0.760 | 0.444 | 0.620 | 0.622 |

edges. It can be observed that the evaluation performance is sensitive to the knowledge context sampled from the graph. A conservative sampling strategy ($\alpha = 1$, $\beta = 1$) will limit the utility of the knowledge in the graph, resulting in a performance closer to baselines IO (see Table 1). On the contrary, an excessively unrestricted sampling ($\alpha = 3$, $\beta = 3$) will result in more irrelevant information and longer input length, thereby deteriorating the performance. Therefore, we practically set $\alpha = 1$ and $\beta = 3$ in all other experiments for convenience.

### D.3 NLI METHOD

Table 6: Comparison of prompt and NLI model for semantic relationship prediction.

| Dataset | LLM | Method | Sentence-wise | | | Response-wise | | |
|---|---|---|---|---|---|---|---|---|
| | | | F1 | F2 | AUC | F1 | F2 | AUC |
| MedHallu-zh | Qwen | LLM prompt | **0.269** | 0.363 | **0.809** | **0.474** | **0.643** | **0.671** |
| | | NLI model | **0.269** | **0.367** | 0.794 | 0.469 | 0.640 | 0.664 |
| | ChatGLM | LLM prompt | **0.228** | **0.360** | **0.798** | 0.445 | 0.634 | 0.622 |
| | | NLI model | 0.227 | 0.356 | 0.793 | **0.452** | **0.640** | **0.625** |
| MedHallu-en | Qwen | LLM prompt | **0.242** | **0.362** | **0.803** | **0.462** | **0.645** | **0.655** |
| | | NLI model | 0.237 | 0.358 | 0.789 | 0.455 | 0.640 | 0.647 |
| | Llama | LLM prompt | **0.180** | **0.296** | **0.747** | **0.408** | **0.628** | **0.581** |
| | | NLI model | 0.179 | 0.289 | 0.746 | 0.397 | 0.627 | 0.572 |

We compare two different methods to predict the semantic relationship between two statements having identical entities: LLM prompts or specific pre-trained NLI models. For LLM prompts, we use prompt shown in Figure 9 and for NLI models, we use StructBERT[8] for MedHallu-zh and DeBERTa[9] for MedHallu-en and WikiBio. The results are listed in Table 6. It can be observed that using prompts consistently performs better than using specific NLI models. However, the differences are trivial and therefore we decided to use prompts in our implementation for convenience.

## E SUPPLEMENTARY RESULTS

### E.1 DETAILED ABLATION RESULTS

Table 7 shows the detailed ablation results.

### E.2 MORE RESULTS OF INFERENCE COSTS

Table 8 shows the more results on inference costs.

## F PROMPTS

---

[8] https://modelscope.cn/models/iic/nlp_structbert_nli_chinese-large
[9] https://huggingface.co/microsoft/deberta-large-mnli

Table 7: Detailed ablation metrics of all variants.

| Dataset | LLM | Variant | Sentence-wise | | | Response-wise | | |
|---|---|---|---|---|---|---|---|---|
| | | | F1 | F2 | AUC | F1 | F2 | AUC |
| MedHallu-zh | Qwen | w/o context | 0.251 | 0.362 | 0.743 | 0.443 | 0.620 | 0.621 |
| | | w/o elicit | 0.252 | 0.362 | 0.796 | 0.440 | 0.633 | 0.615 |
| | | w/o sample | 0.262 | **0.389** | **0.812** | 0.454 | 0.637 | 0.651 |
| | | w/o conflict | 0.265 | 0.359 | 0.808 | 0.469 | 0.642 | 0.667 |
| | | full | **0.269** | 0.363 | 0.809 | **0.474** | **0.643** | **0.671** |
| | ChatGLM | w/o context | 0.211 | 0.355 | 0.787 | 0.442 | 0.635 | 0.613 |
| | | w/o elicit | 0.214 | 0.357 | 0.789 | 0.442 | **0.644** | 0.613 |
| | | w/o sample | 0.215 | 0.331 | 0.774 | 0.429 | 0.629 | 0.619 |
| | | w/o conflict | 0.211 | 0.349 | 0.787 | 0.440 | 0.636 | 0.610 |
| | | full | **0.228** | **0.360** | **0.798** | **0.445** | 0.634 | **0.622** |
| MedHallu-en | Qwen | w/o context | 0.231 | 0.355 | 0.794 | 0.440 | 0.628 | 0.621 |
| | | w/o elicit | 0.234 | 0.357 | 0.800 | 0.442 | 0.624 | 0.623 |
| | | w/o sample | 0.230 | **0.365** | 0.802 | 0.433 | 0.638 | 0.627 |
| | | w/o conflict | 0.226 | 0.301 | 0.671 | 0.401 | 0.620 | 0.572 |
| | | full | **0.242** | 0.362 | **0.803** | **0.462** | **0.645** | **0.655** |
| | Llama | w/o context | 0.145 | 0.277 | 0.702 | **0.419** | **0.636** | **0.585** |
| | | w/o elicit | 0.170 | 0.284 | 0.727 | 0.408 | 0.629 | 0.569 |
| | | w/o sample | 0.170 | 0.281 | 0.726 | 0.404 | 0.627 | 0.563 |
| | | w/o conflict | 0.169 | 0.268 | 0.668 | 0.406 | 0.620 | 0.569 |
| | | full | **0.180** | **0.296** | **0.747** | 0.408 | 0.628 | 0.581 |

Table 8: Inference costs of ChatGLM and Llama.

| Dataset | Method | Relative Perform.↑ | #Call↓ | Relative #Call↓ | #Token↓ (k) | Relative #Token↓ |
|---|---|---|---|---|---|---|
| ChatGLM | IO | -4.6% | 7552 | -55.3% | 144 | -87.8% |
| | ContextIO | -7.3% | 7552 | -55.3% | 119 | -89.9% |
| | HistoryIO | -1.7% | 7552 | -55.3% | 191 | -83.8% |
| | CoT | -23.2% | 7,552 | -55.3% | 505 | -57.0% |
| | CoVE | -13.6% | 35,152 | +108.0% | 1,318 | +12.1% |
| | FaR | -11.6% | 14,104 | -16.5% | 1,550 | +32.0% |
| | SelfCheckGPT | -30.0% | 135,472 | +701.6% | 8,710 | +641.3% |
| | ChatProtect | -17.7% | 115,144 | +581.3% | 2,291 | +95.0% |
| | SelfElicit | - | 16,901 | - | 1,175 | - |
| Llama | IO | -5.6% | 7,422 | -34.9% | 526 | -60.1% |
| | ContextIO | -9.1% | 7,422 | -34.9% | 718 | -45.6% |
| | HistoryIO | -8.0% | 7,422 | -34.9% | 449 | -66.0% |
| | CoT | -14.9% | 7,422 | -34.9% | 1,285 | -2.7% |
| | CoVE | -25.1% | 35,222 | +208.7% | 3,277 | +148.2% |
| | FaR | -15.8% | 14,104 | +23.6% | 4,030 | +205.1% |
| | SelfCheckGPT | -16.7% | 130,912 | +1047.4% | 17,913 | +1256.4% |
| | ChatProtect | -15.9% | 148,618 | +1202.6% | 12,560 | +851.1% |
| | SelfElicit | - | 11,409 | - | 1,321 | - |

---

### Prompt for identifying named entities and extracting knowledge statements

You are a knowledge extractor. Your task is to identify named entities from the given sentences and extract the knowledge points related to these entities.
Steps:
1. For each sentence, identify the named entities within. Named entities include, but are not limited to: {{entity types}}
   Please use the format "Named entities in sentence 1: Entity 1 (Type 1)" to list all the named entities you find.
2. For each identified named entity, extract all the related knowledge points, ensuring the semantic integrity of the points, and that they can be understood independently from the original sentence. If independent knowledge points cannot be extracted, please return the original sentence directly. Please use the format "Knowledge points in sentence 1: [Knowledge point 1][Knowledge point 2]"to list all the knowledge points you find.
{{few shot}}
Your task is to provide named entities and knowledge points based on the following sentence:
{{sentence}}
Named entities:

Figure 8: Prompt for identifying named entities and extracting knowledge statements in Section 3.1.

---

### Prompt for detecting the relation between two statements

Please determine the semantic relationship between the following two sentences. There are three possible types of relationships:
1. [entail]: The content of the two sentences is the identical, describing the same aspect of the same object, with consistent content.
2. [contradict]: The two sentences describe the same aspect of the same object, but the content is directly opposite, presenting a contradiction.
3. [neutral]: The two sentences describe different objects, or different aspects of the same object, and can coexist.
Please analyze sentence A and sentence B, and choose one of the relationships. Please briefly explain your reasoning.
Sentence A: {{SENTENCE_A}}
Sentence B: {{SENTENCE_B}}
Judgment result:

Figure 9: Prompt for detecting the relation between two statements.

**Prompt for mitigating the conflicts between two statements**

Please read the following two sentences.
These two sentences describe the same aspect of the same object, but their content is contradictory.
Your task is to judge which sentence is more accurate based on your own understanding.
Sentence A: {{SENTENCE_A}}
Sentence B: {{SENTENCE_B}}
Judging criteria:
Please consider the logic and factual basis of the sentences. Choose the sentence you think is correct and select from the following two options:
[Sentence A is correct]
[Sentence B is correct]

Figure 10: Prompt for mitigating the conflicts between two statements.

**Prompt for identifying check-worthy sentences with domain expertise**

You will be handling questions and answers related to medical consultations and healthcare. Your task is to categorize a sentence from the response based on its content. Classify the sentence accurately under one of the following categories:
1. [Medical Knowledge]: Includes objective descriptions of medical knowledge, detailing specific diseases, symptoms, medications, methods, etc. Examples include:
   a. Ezetimibe is a cholesterol absorption inhibitor that reduces cholesterol absorption in the gut, thereby lowering blood lipids
   ....
2. [Personal Condition]: Describes the current state of a specific patient (complaints, history, laboratory data, signs), without including treatment or advice. Examples include:
   a.  Age 48, tumor marker carcinoembryonic antigen 100
   ....
3. [Lifestyle]: Discusses health and lifestyle habits other than treatment. Examples include:
   a.  Increasing physical exercise can effectively reduce the risk of cardiovascular disease
   ....
4. [Other]: Sentences that do not fit into any of the above categories, such as emotional expression type, subjective evaluation type, non-medical type, etc.
Please identify which category the following sentence from the response belongs to:
{{sentence}}

Figure 11: Prompt for identifying check-worthy sentences with domain expertise for MedHalluZH and MedHalluEN.

## Prompt for translating samples

===============================User===============================

Please translate the following sentences about medicine into English. Only output the translated sentences with serial numbers and nothing else. Sentences to be translated into English:
1. {{sentence 1}}
2. {{sentence 2}}
3. {{sentence 3}}
...
Please translate:

Figure 12: Prompt for translating Chinese dataset into English version.

## Prompt for labeling sentences according to the experts' comment

============================Instructions:============================

1. You are given several sentences and a comment. The comment points out the incorrectness of some of the sentences. Your task is to find the incorrect sentence pointed out by the comment.
2. The sentences are given in a list. Each sentence starts with "- ".
3. The comment might include satisfaction issues, correctness issues, and universal issues. But you should only focus on the correctness issues.
4. Find the incorrect sentences pointed out by the comment. Note that in some cases all sentences might be correct and there is no incorrect sentence.
5. You should only copy the incorrect sentences as a list, with each item starting with "- ". Do not include other formatting. If there is no incorrect sentence, reply "- ".
6. The sentences are annotated with <sentence>. The comment is annotated with <comment>. Your task is to do this for the given <sentence> and <comment>.

<sentence>
- {{sentence 1}}
- {{sentence 2}}
<comment>
{{comment}}

Figure 13: Prompt for LLM-aided labeling.

