# OpenReview forum: "Long-form Hallucination Detection with Self-elicitation"
_ICLR.cc/2025/Conference — ICLR 2025 Conference Withdrawn Submission_

### Official Review · Reviewer_eFCV · 2024-10-27

**Soundness:** 2
**Presentation:** 3
**Contribution:** 3
**Rating:** 6
**Confidence:** 3

**Summary:**

The paper presents an integrated long-form hallucination detection mechanism, SelfElicit, which centers around a knowledge hypergraph and includes a series of introspective steps, allowing hallucination detection without the need for external knowledge retrieval. This approach demonstrates academic originality and effectively expands the applicability of hallucination detection. The paper is well-structured, employs precise language, and demonstrates rigorous logic. Extensive experiments validate the effectiveness of the proposed SelfElicit framework, and quantitative analyses further investigate the quality of insights generated by SelfElicit. I believe this paper will make a meaningful contribution to the field.

**Strengths:**

1.The paper introduces an innovative long-form hallucination detection mechanism, SelfElicit, which synergizes the self-elicitability of inherent knowledge within large language models with long-form continuity comprehension. This method skillfully leverages the concept of self-generation within large models to elicit expressions of their intrinsic knowledge.

2.The paper is well-structured, with a clear description of the methodology and a comprehensive experimental section.

3.This work expands the applicability of hallucination detection through SelfElicit, potentially contributing meaningfully to the research community.

**Weaknesses:**

1. Some viewpoints in the paper lack factual support. For instance, in line 214, the statement "prompting the model to provide reflections on the evaluation" lacks evidence or justification.

2. Although mentioned in the Limitations section, it is worth emphasizing that the experiments are conducted solely on real-world medical QA datasets. This limitation narrows the scope of general conclusions and reduces the credibility of SelfElicit’s generalizability across diverse fields.

3. In the Main Results section, the baseline is locally implemented rather than based on data from previous research, which diminishes the persuasive strength of the experimental findings.

**Questions:**

1. Why is the "prompting the model to provide reflections" approach a more effective method? This actually corresponds precisely to weaknesses 1.

2. When handling statements categorized as "Contradict" the model still relies on its own judgment to identify conflicts and make corrections. Does this approach lead to cumulative errors? In other words, as the framework operates through its complete chain, it continuously encounters potential hallucinations. How do you view the cumulative error introduced by this process?

3. In this workflow, the knowledge hypergraph provides a stable limitation on LLM hallucinations. However, during updates, the hypergraph itself might become contaminated with hallucinated information. Notably, in the w/o conflict setting, the performance decline is substantial. Since the article only briefly addresses this issue in line 257, I would like to understand further how this contamination is mitigated.

4. In the proposed self-elicit method, is the core mechanism based more on the model's intrinsic knowledge expression or the knowledge hypergraph?

5. In the Main Results section, the baseline was implemented locally rather than using data from other studies, which may weaken the persuasive power of the experimental results. Could you provide comparative data from similar scenarios in other literature to strengthen credibility? (For example, using results from SelfCheckGPT under comparable experimental conditions).

---

> ### Author Response · Authors · 2024-11-20
> **Response to Reviewer eFCV (part 1)**
>
> We sincerely thank the reviewers for the constructive feedback.
>
> ## Weakness 1 & Question 1
>
> >  Some viewpoints in the paper lack factual support. For instance, in line 214, the statement "prompting the model to provide reflections on the evaluation" lacks evidence or justification.
>
> > Why is the "prompting the model to provide reflections" approach a more effective method? This actually corresponds precisely to weaknesses 1\.
>
> We acknowledge that it might not be the optimal or most effective. The question seems to arise from a misunderstanding. In the manuscript, we believe that “..is a more **convenient** method”, since it directly follows the thoughts of fact-checking and does not require extra steps to elicit the knowledge. Therefore, it is technically more simple.
>
> Moreover, in the experiments, we compare baselines with various elicitation methods, including self-ask (CoVE), self-consistency (SelfCheckGPT and ChatProtect), and chain-of-thought (CoT and FaR). Compared with these methods, SelfElicit utilized reflections conditioned on the evaluation, which shows better factuality and diversity (Section 4.3 Elicitation Quality).
>
> ## Weakness 2
>
> >  Although mentioned in the Limitations section, it is worth emphasizing that the experiments are conducted solely on real-world medical QA datasets. This limitation narrows the scope of general conclusions and reduces the credibility of SelfElicit’s generalizability across diverse fields.
>
> We have conducted experiments on the WikiBio\[1\] dataset, which includes the biographies of 238 celebrities. In summary, the result shows that SelfElicit has strong performance across different domains. Detailed metrics and discussions can be found in our response to Weaknesses 8 & 11, Reviewer LceZ (link: https://openreview.net/forum?id=r9mYbs8RTH&noteId=ttwwA4UQJX).
>
> ## Weakness 3 & Question 5
>
> > In the Main Results section, the baseline is locally implemented rather than based on data from previous research, which diminishes the persuasive strength of the experimental findings.
>
> > In the Main Results section, the baseline was implemented locally rather than using data from other studies, which may weaken the persuasive power of the experimental results. Could you provide comparative data from similar scenarios in other literature to strengthen credibility? (For example, using results from SelfCheckGPT under comparable experimental conditions).
>
> We sincerely thank you for the suggestions. However, due to various realistic limitations, we have to implement these methods locally, which include: (1) using private datasets (CoVE), and (2) focusing on short-form datasets (IO, ChatProtect, and FaR).
>
> More importantly, the main focus of the experiment is to compare the effectiveness of various elicitation methods. Therefore, we use the *identical self-eval prompt* after their original procedures to obtain the hallucination scores for a fair comparison. In other words, the only difference between all methods is the approach to elicit the intrinsic knowledge. Unfortunately, we have to implement these baselines locally rather than using metrics reported in their papers since they have used different methods to obtain the hallucination score.

---

> > ### Author Response · Authors · 2024-11-20
> > **Response to Reviewer eFCV (part 2)**
> >
> > ## Questions 2 & 3
> >
> > > When handling statements categorized as "Contradict" the model still relies on its own judgment to identify conflicts and make corrections. Does this approach lead to cumulative errors? In other words, as the framework operates through its complete chain, it continuously encounters potential hallucinations. How do you view the cumulative error introduced by this process?
> >
> > >  In this workflow, the knowledge hypergraph provides a stable limitation on LLM hallucinations. However, during updates, the hypergraph itself might become contaminated with hallucinated information. Notably, in the w/o conflict setting, the performance decline is substantial. Since the article only briefly addresses this issue in line 257, I would like to understand further how this contamination is mitigated.
> >
> > Thank you for providing this suggestion.
> >
> > Theoretically, we acknowledge the iterative chain has an underlying risk of cumulative errors when the NLI process is not activated or fails to detect inconsistencies (detailed failure modes are discussed in Weakness 12, Reviewer LceZ, link: https://openreview.net/forum?id=r9mYbs8RTH&noteId=aTyDKFIFY7).
> >
> > We argue that the graph sampling and conflict detection/mitigation procedures can partially alleviate this issue by filtering out irrelevant information and discarding incorrect reflections via self-consistency. The conflict detection/mitigation procedures have the capability of removing contamination already existing in the hypergraph by replacing it with new reflections that contradict the contamination. In Section 4.5 in the revised version, we show a case study to demonstrate how the inaccuracy is mitigated via conflict detection.
> >
> > We acknowledge this is an underlying limitation of our method currently. In future works, we will consider several possible approaches such as introducing external databases or adding extra validation for better credibility.
> >
> > ## Question 4
> >
> > >  In the proposed self-elicit method, is the core mechanism based more on the model's intrinsic knowledge expression or the knowledge hypergraph?
> >
> > Intrinsic knowledge expression. The core fact-checking is based on the intrinsic knowledge expression (② in Figure 2). The knowledge hypergraph is an auxiliary “memory” to store the expressed intrinsic knowledge.
> >
> > Experimentally, without the knowledge hypergraph (discarding the sampling, conflict detection, and merging procedures), SelfElicit degrades into HistoryIO, which still achieves better performance than most baselines, showing that intrinsic knowledge expression brings the most performance gain.
> >
> > ## References
> >
> > \[1\]WikiBio. [https://huggingface.co/datasets/potsawee/wiki\_bio\_gpt3\_hallucination](https://huggingface.co/datasets/potsawee/wiki_bio_gpt3_hallucination)
> >
> > \[2\]SelfCheckGPT: Zero-Resource Black-Box Hallucination Detection for Generative Large Language Models

---

> > > ### Comment · Reviewer_eFCV · 2024-11-21
> > > **Response to authors**
> > >
> > > Thank you for the detailed reply. My concerns have been largely addressed, and the weaknesses I previously mentioned have been effectively addressed as well.
> > >
> > > Additionally, I noticed some suggestions from other reviewers regarding writing style. Adopting a more precise and rigorous tone could make the paper even more suitable for publication at ICLR.
> > >
> > > I believe it is appropriate to increase both the contribution and rating.

---

> > > > ### Author Response · Authors · 2024-11-22
> > > > **Response to Reviewer eFCV**
> > > >
> > > > Thanks again for your review!

---

### Official Review · Reviewer_GtBE · 2024-11-01

**Soundness:** 2
**Presentation:** 3
**Contribution:** 2
**Rating:** 5
**Confidence:** 4

**Summary:**

The paper "Long-Form Hallucination Detection with Self-Elicitation" presents a novel approach to address the issue of factual inaccuracies, or hallucinations, in the output of Large Language Models (LLMs), especially in the context of long-form text generation. The authors introduce SelfElicit, a framework that leverages the intrinsic knowledge of LLMs and the semantic continuity within long-form content to effectively detect and mitigate hallucinations. The key contributions include a self-elicitation mechanism to elicit the models' knowledge, the use of a knowledge hypergraph for organized knowledge retention and inconsistency resolution, and the demonstration of the framework's effectiveness through extensive experiments on real-world medical QA datasets. The paper claims that SelfElicit outperforms existing methods in detecting hallucinations in long-form content, offering a significant step towards improving the reliability of LLM-generated text.

**Strengths:**

1. The paper uniquely addresses the semantic continuity in long-form content, which is often overlooked, and shows how this can improve the detection of factual inaccuracies.
2. The use of a knowledge hypergraph for organizing and retaining knowledge is a significant strength, as it helps in managing complexity and resolving inconsistencies in the model's knowledge base.
3. The paper clearly identifies the problem of hallucination in LLMs and provides a well-aligned solution, making it easy for readers to follow the rationale behind the proposed framework and its components.

**Weaknesses:**

1. The paper's focus on medical QA datasets raises concerns about the framework's ability to generalize to other domains. The lack of experimentation in diverse domains means that the robustness and applicability of SelfElicit across different types of long-form content remain unproven.
2. The theoretical underpinnings of the self-elicitation process and its interaction with the knowledge hypergraph are not thoroughly explored. The paper does not provide a deep dive into the mathematical or conceptual models that would support the claims about the framework's effectiveness.
3. The paper heavily relies on quantitative metrics to evaluate the performance of the SelfElicit framework. However, there is an absence of qualitative analysis, such as case studies or detailed error analysis, which could provide a more nuanced understanding of the framework's strengths and weaknesses in practical scenarios.
4. The visual representation in the paper, particularly in Figure 2, is not sufficiently clear or expressive, which can lead to confusion about the SelfElicit framework's methodology. The diagrams fail to effectively communicate the process, detracting from the reader's understanding and the overall quality of the presentation. A redesign of these figures is recommended for better clarity.
5. The paper claims that SelfElicit outperforms existing methods but does not provide a detailed analysis or comparison with SOTA techniques. The superficial nature of these comparisons leaves room for doubt regarding the true novelty and superiority of the proposed framework.

**Questions:**

1. How well does the SelfElicit framework generalize to different domains beyond medical QA? Have there been any preliminary tests or simulations to assess its performance on long-form content from other fields, such as legal or historical texts?
2. What is the theoretical foundation that underpins the self-elicitation mechanism and its interaction with the knowledge hypergraph? Could the authors provide more details on any formal proofs or mathematical models that validate the effectiveness of this approach?
3. What are the scalability considerations of the SelfElicit framework? How does its performance and accuracy change with increasingly larger and more complex datasets, and are there any optimizations in place to handle such scalability challenges?

---

> ### Author Response · Authors · 2024-11-20
> **Response to Reviewer GtBE (part 1)**
>
> We sincerely thank the reviewers for the constructive feedback.
>
> ## Weakness 1 & Question 1
>
> >  Weaknesses 1: The paper's focus on medical QA datasets raises concerns about the framework's ability to generalize to other domains. The lack of experimentation in diverse domains means that the robustness and applicability of SelfElicit across different types of long-form content remain unproven.
>
> > Question 1: How well does the SelfElicit framework generalize to different domains beyond medical QA? Have there been any preliminary tests or simulations to assess its performance on long-form content from other fields, such as legal or historical texts?
>
> We have conducted experiments on the WikiBio\[1\] dataset, which includes the biographies of 238 celebrities. In summary, the result shows that SelfElicit has strong performance across different domains. Detailed metrics and discussions can be found in our response to Weaknesses 8 & 11, Response to Reviewer LceZ (part 3), link: https://openreview.net/forum?id=r9mYbs8RTH&noteId=ttwwA4UQJX).
>
> To the best of our knowledge, there are currently no publicly available datasets from other domains. And we will leave the work of synthesizing multi-domain data to future works due to time limits.
>
> ## Weakness 2 & Question 2
>
> > The theoretical underpinnings of the self-elicitation process and its interaction with the knowledge hypergraph are not thoroughly explored. The paper does not provide a deep dive into the mathematical or conceptual models that would support the claims about the framework's effectiveness.
>
> > What is the theoretical foundation that underpins the self-elicitation mechanism and its interaction with the knowledge hypergraph? Could the authors provide more details on any formal proofs or mathematical models that validate the effectiveness of this approach?
>
> **self-elicitation**
>
> We acknowledge that the motivation of self-elication is largely experimental and we have not yet fully understand the mathematical model. However, we have some conceptual understandings about the effectiveness.
>
> Conceptually, SelfElicit has schematic similarities with multi-step reasoning. Since LLMs face difficulties in using their stored knowledge to perform multi-step inference tasks\[4\], our SelfElicit alleviates these issues in two aspects. On one hand, the model only focuses on one specific factoid at a time, which makes it easier to utilize their intrinsic knowledge. More importantly, *since parts of the intrinsic knowledge have been expressed verbally in advance, the reasoning burden can be reduced with the help of the in-context information*. Similar gains can also be found in a boarder context\[4\]\[5\]\[6\]\[7\]. In this work, the capacity of multi-step reasoning is integrated with the calibration capacity of LLMs for effective long-form hallucination detection.
>
> **knowledge hypergraph**
>
> The interaction with the knowledge hypergraph is for the **technical** convenience of knowledge organization. Compared with HistoryIO (prefixing historical query-responses as contextual information), SelfElicit further uses graph structures to enable a more careful organization of the elicited information via sampling, conflict detection, and mitigation, leading to better performance. Moreover, graphs provide more intuitive knowledge organization, better efficiency, and higher underlying expansibility than text-based storage.
>
> ## Weakness 3
>
> > The paper heavily relies on quantitative metrics to evaluate the performance of the SelfElicit framework. However, there is an absence of qualitative analysis, such as case studies or detailed error analysis, which could provide a more nuanced understanding of the framework's strengths and weaknesses in practical scenarios.
>
> We have added two case studies in Section 4.5 in the revised version to demonstrate the workflow of our method. One is an evaluation with/without contextual information, showing the effectiveness of self-elicited thoughts, and the other is a model generating inaccurate reflection and how it is mitigated by conflict detection.
>
> ## Weakness 4
>
> > The visual representation in the paper, particularly in Figure 2, is not sufficiently clear or expressive, which can lead to confusion about the SelfElicit framework's methodology. The diagrams fail to effectively communicate the process, detracting from the reader's understanding and the overall quality of the presentation. A redesign of these figures is recommended for better clarity.
>
> We have redesigned Figure 2 to make it clearer, including a reorganized diagram, clearer arrows, and example texts. We have also included a case study with a detailed data case in Section 4.5 in the revised version.

---

> > ### Author Response · Authors · 2024-11-20
> > **Response to Reviewer GtBE (part 2)**
> >
> > ## Weakness 5
> >
> > > The paper claims that SelfElicit outperforms existing methods but does not provide a detailed analysis or comparison with SOTA techniques. The superficial nature of these comparisons leaves room for doubt regarding the true novelty and superiority of the proposed framework.
> >
> > We apologize for the confusion regarding the choice of the baselines and have re-categorized the baselines to highlight their characteristics:
> >
> > * Classic self-evaluation (IO): Probability of the False token following a query whether the statement is factual.
> > * Long-form argument approaches
> >   * ContextIO: *Prior evaluated statements are prefixed as contextual information* before evaluation.
> >   * HistoryIO: *Historical information (queries and responses) of prior evaluations are prefixed as contextual information* before evaluation.
> > * Various elicitation approaches
> >   * Chain-of-thought (CoT, FaR): *Elicitation by step-by-step reasoning* before evaluation.
> >   * self-ask (CoVE): *Elicitation by answering fact-related questions generated by the model itself* before evaluation.
> >   * self-consistency (SelfCheckGPT, ChatProtect): *Elicitation by answering multi-aspect questions and measuring the consistency* before evaluation.
> >
> > For a fair comparison, we use the identical self-eval prompt (IO) after their original procedures to obtain the hallucination scores. In other words, the only difference between all methods is the approach to elicit the intrinsic knowledge.
> >
> > The results in Table 1 show that SelfElicit achieves superior performance, which demonstrates the effectiveness of the self-elicitation mechanism against existing methods (chain-of-thought, self-ask, and self-consistency).
> >
> > We have accordingly updated Table 1 and the introduction of baselines.
> >
> > ## Question 3
> >
> > >  What are the scalability considerations of the SelfElicit framework? How does its performance and accuracy change with increasingly larger and more complex datasets, and are there any optimizations in place to handle such scalability challenges?
> >
> > Since the evaluation of each sample is independent, we consider the cost of fact-checking a single sample in the following discussion.
> >
> > We denote the number of factoids in a sample as $F$. The complexities of each component are:
> >
> > * **sampling & merging**: rule-based, no LLM usage. Since the sampling is merely edge retrieval rather than graph traversal, the overhead is trivial.
> > * **evaluation & reflection**: $\mathcal{O}(F)$ LLM calls in total to evaluate all statements one by one.
> > * **conflict detection**: in extreme cases, each new edge is conflicted with the existing ones in the graph. Therefore, the complexity is $\mathcal{O}(F^2)$.
> > * **conflict mitigation**. Similarly, the complexity is $\mathcal{O}(F^2)$.
> >
> > In summary, the overall complexity is $\mathcal{O}(F^2)$ for each input sample, which indicates an underlying quadratic increase of overhead as the number of factoids of a sample as $F$ increases. We acknowledge it is a potential limitation of our work.
> >
> > However, we believe that this scalability issue is not the major challenge in current hallucination detection scenarios. On one hand, the conflict detection/mitigation procedures are not always activated (the detection is activated in 17.76% (801/4510) cases and the mitigation is activated in 1.06%(48/4510) cases, qwen1.5-7B in MedHalluEN). On the other hand, the scales of $F$ in current datasets are not large enough to raise major issues (MedHallu: avg/max values are 7.37/26, WikiBio: avg/max values are 8.01/13).
> >
> > To handle potential dataset scalability challenges in the future, several in-place optimizations can be made: (1) improving the graph sampling procedure to better filter out irrelevant information, (2) improving the detection approaches for better inconsistency detection (e.g. traversal \+ embedding-based methods like BM25\[2\]), and (3) adapting graph clustering approach\[3\] to reduce the scale of the graph.
> >
> > ## References
> >
> > \[1\]WikiBio. [https://huggingface.co/datasets/potsawee/wiki\_bio\_gpt3\_hallucination](https://huggingface.co/datasets/potsawee/wiki_bio_gpt3_hallucination)
> >
> > \[2\]Some simple effective approximations to the 2-poisson model for probabilistic weighted retrieval.1994.
> >
> > \[3\]From Local to Global: A Graph RAG Approach to Query-Focused Summarization. 2024\.
> >
> > \[4\]Iteratively Prompt Pre-trained Language Models for Chain of Thought. 2022\.
> >
> > \[5\]Chain-of-Thought Prompting Elicits Reasoning in Large Language Models. 2022\.
> >
> > \[6\]STaR: Bootstrapping Reasoning With Reasoning.
> >
> > \[7\]Prompted LLMs as Chatbot Modules for Long Open-domain Conversation.2023.

---

> ### Comment · Reviewer_GtBE · 2024-11-25
>
> Thank you for your reply and addressed some concerns, so I will improve my scores.

---

### Official Review · Reviewer_bJeQ · 2024-11-01

**Soundness:** 2
**Presentation:** 3
**Contribution:** 3
**Rating:** 6
**Confidence:** 4

**Summary:**

The paper presents a novel framework called SelfElicit for detecting hallucinations in long-form content generated by Large Language Models (LLMs). The approach leverages self-elicitation of intrinsic knowledge and understanding of semantic continuity to improve hallucination detection.

**Strengths:**

Improved Hallucination Detection: The framework effectively detects hallucinations by leveraging self-elicitation and semantic continuity, resulting in superior performance compared to existing methods.

Intrinsic Knowledge Utilization: By eliciting intrinsic knowledge from LLMs, the approach enhances the factuality and diversity of knowledge expression.

Graph-Based Contextual Understanding: The use of a knowledge hypergraph allows for better organization and understanding of semantic continuity, aiding in more accurate evaluations.

Iterative Improvement: The framework iteratively refines its understanding and knowledge base, potentially leading to more accurate and reliable results over time.

**Weaknesses:**

Complexity: The approach involves multiple steps and components, which may increase the complexity of implementation and computational overhead. To better understand the scalability of the framework, could the authors provide a detailed analysis of the computational complexity? Additionally, it would be helpful to see comparisons with existing methods in terms of implementation complexity and runtime. Not so sure about scalability here.

Dependency on LLMs: The performance of the framework is inherently tied to the capabilities of the underlying LLMs, which may limit its effectiveness in certain scenarios. Could the authors provide a more detailed analysis of how the framework's performance varies with different LLMs? It would also be beneficial to include a comparison with non-LLM based approaches and a cost-benefit analysis that weighs the improved performance against potential increased costs.

Potential for Induced Hallucinations: While the framework aims to mitigate hallucinations, the process of self-elicitation and reflection may still introduce inaccuracies or hallucinations, especially with ambiguous or unfamiliar content. Have the authors considered implementing additional safeguards or validation steps to catch induced hallucinations? Specific strategies or examples would be helpful in understanding how this limitation might be addressed.

**Questions:**

How does the framework handle cases where the LLM lacks sufficient intrinsic knowledge about a specific topic?

What are the computational costs associated with maintaining and updating the knowledge hypergraph, especially for large datasets? Could the authors provide specific metrics or comparisons, such as how these costs scale with the size of the input or the number of iterations?

How does the framework handle cases where the LLM lacks sufficient intrinsic knowledge about a specific topic? Have the authors tested the framework's performance on topics known to be outside the LLM's training data? Additionally, is there a fallback mechanism in place for when the LLM expresses low confidence?

Can the approach be adapted or extended to work with external knowledge sources to further enhance hallucination detection?

How does the framework perform across different domains and languages, and what adaptations might be necessary for domain-specific applications?

---

> ### Author Response · Authors · 2024-11-20
> **Response to Reviewer bJeQ (part 1)**
>
> ## Weakness 1
>
> > Complexity: The approach involves multiple steps and components, which may increase the complexity of implementation and computational overhead. To better understand the scalability of the framework, could the authors provide a detailed analysis of the computational complexity? Additionally, it would be helpful to see comparisons with existing methods in terms of implementation complexity and runtime. Not so sure about scalability here.
>
> Since the evaluation of each sample is independent, the discussion is under the condition when fact-checking a single sample.
>
> Assume there are $N$ statements (or $F$ factoids) to be fact-checked in a given sample. We categorize the LLM usages into two categories: **token** where only the first token or its logit matters, and **open-ended** where LLMs generate full reasoning given an instruction. Note that an open-ended generation will be magnitudes more costly than a token generation because the former usually generates hundreds of tokens while the latter generates only several tokens.
>
> Table 1\. Complexity analysis of all methods. $N$: the number of sentences (average 6.81). $F$: the number of factoids (average 7.37). $K$: the number of generated documents. Brackets denote the components that require LLM usage.
>
> | method       | token                                                      | open-ended                                |
> | :----------- | :--------------------------------------------------------- | :---------------------------------------- |
> | IO           | N(eval)                                                    | \-                                        |
> | ContextIO    | N(eval)                                                    | \-                                        |
> | HistoryIO    | N(eval)                                                    | \-                                        |
> | CoT          | N(eval)                                                    | N(reasoning)                              |
> | CoVE         | FN(result comparison)                                      | N(question generation) \+ FN(answer)      |
> | FaR          | N(eval)                                                    | N(elicit) \+ N(reflect)                   |
> | SelfCheckGPT | KN(eval)                                                   | K(document generation)                    |
> | ChatProtect  | FN(eval)                                                   | N(extract triple) \+ FN(cloze the triple) |
> | SelfElicit   | F(eval) \+ F(conflict detection) \+ F(conflict mitigation) | 1(extract) \+ F(reflection)               |
>
> Since an open-ended generation is magnitudes more costly than a token generation, the major overhead is the open-ended generation. In summary, SelfElicit (1+F) has a comparable complexity with CoT (N) and is faster than CoVE (N+FN), FaR (N+N), and ChatProtect (N+FN), which is also demonstrated by the experimental results in Table 2 in Section 5.1.
>
> ## Weakness 2.1
>
> >  Dependency on LLMs: The performance of the framework is inherently tied to the capabilities of the underlying LLMs, which may limit its effectiveness in certain scenarios. Could the authors provide a more detailed analysis of how the framework's performance varies with different LLMs?
>
> The performance is inherently tied to the capabilities of the underlying LLMs. In Table 2, we compare two families of modern LLMs: the qwen family (qwen1.5 against qwen2.5) and the Llama family (llama2 against llama3.1) with close model scales.
>
> (1) Llama3.1 and Qwen2.5 outperform their previous-generation counterparts in almost all cases, respectively, which is owe to their stronger capability.
>
> (2) In some cases, with appropriate algorithms, the performance of previous-generation models surpasses latest-generation models (e.g. SelfElicit+Llama2 \> IO+Llama3). This phenomenon shows that appropriate algorithms can benefit fact-checking and boost performance without upgrading the LLMs.
>
> (3) SelfElicit beats all other baselines across different LLM backbones in most cases, which demonstrates the effectiveness of our method.

---

> > ### Author Response · Authors · 2024-11-20
> > **Response to Reviewer bJeQ (part 2)**
> >
> > Table 2: Qwen/Qwen2 and Llama2/Llama3 comparison on MedHallu-EN. S: sentence-level metrics. R: response-level metrics. **Bold**: better than counterparts.
> >
> > | Methods |      | SelfElicit |           |    IO     |           | ContextIO |           | HistoryIO |           |    CoT    |           |   CoVE    |           |    FaR    |           | SelfChkGPT |           | ChatProtect |           |
> > | :-----: | :--: | :--------: | :-------: | :-------: | :-------: | :-------: | :-------: | :-------: | :-------: | :-------: | :-------: | :-------: | :-------: | :-------: | :-------: | :--------: | :-------: | :---------: | :-------: |
> > | Metric  |      |    *F1*    |   *AUC*   |   *F1*    |   *AUC*   |   *F1*    |   *AUC*   |   *F1*    |   *AUC*   |   *F1*    |   *AUC*   |   *F1*    |   *AUC*   |   *F1*    |   *AUC*   |    *F1*    |   *AUC*   |    *F1*     |   *AUC*   |
> > |  Qwen   |  S   |   0.242    |   0.803   |   0.182   |   0.762   |   0.168   |   0.743   |   0.233   |   0.779   |   0.192   |   0.596   |   0.085   |   0.500   |   0.187   |   0.763   | **0.226**  | **0.682** |    0.085    |   0.505   |
> > |         |  R   |   0.463    |   0.656   |   0.436   |   0.622   |   0.443   |   0.614   | **0.472** | **0.659** |   0.395   | **0.570** |   0.395   |   0.498   |   0.445   |   0.630   |   0.428    |   0.623   |  **0.395**  |   0.505   |
> > |  Qwen2  |  S   | **0.282**  | **0.820** | **0.275** | **0.805** | **0.247** | **0.802** | **0.254** | **0.811** | **0.211** | **0.636** | **0.259** | **0.672** | **0.217** | **0.784** |   0.232    |   0.675   |  **0.087**  | **0.523** |
> > |         |  R   | **0.479**  | **0.667** | **0.466** | **0.665** | **0.460** | **0.661** |   0.456   |   0.656   | **0.422** |   0.422   | **0.440** | **0.614** | **0.447** | **0.640** | **0.444**  | **0.636** |  **0.395**  | **0.537** |
> > | Llama2  |  S   |   0.181    |   0.748   |   0.137   |   0.697   |   0.139   |   0.705   |   0.133   | **0.667** |   0.142   |   0.594   |   0.085   |   0.499   |   0.140   |   0.709   |   0.103    |   0.561   |    0.136    |   0.550   |
> > |         |  R   |   0.408    |   0.582   | **0.410** | **0.555** | **0.407** |   0.509   | **0.413** |   0.551   |   0.395   |   0.537   |   0.395   |   0.497   |   0.411   |   0.558   |   0.397    |   0.547   |    0.395    |   0.568   |
> > | Llama3  |  S   | **0.211**  | **0.773** | **0.156** | **0.724** | **0.170** | **0.741** |   0.147   |   0.662   | **0.223** | **0.666** | **0.184** | **0.699** | **0.184** | **0.730** | **0.158**  | **0.634** |  **0.208**  | **0.601** |
> > |         |  R   | **0.447**  | **0.622** |   0.406   |   0.546   |   0.405   | **0.572** | **0.413** | **0.605** | **0.449** | **0.626** | **0.421** | **0.562** | **0.422** | **0.586** | **0.417**  | **0.613** |  **0.414**  | **0.600** |
> >
> > Table 3\. Accuracy gain v.s. cost in MedHallu-ZH.
> >
> > | MedHalluZH | accuracy | \#correct | \#incorrect | \#true\_pos | \#false\_pos | \#calls / sample | time / sample(s) |
> > | :--------- | :------- | :-------- | :---------- | :---------- | :----------- | :--------------- | :--------------- |
> > | w/o detect | 0.7537   | 612       | 200         | \-          | \-           | \-               | \-               |
> > | retrieval  | 0.7620   | 583       | 182         | 18          | 29           | \-               | 12.4             |
> > | qwen1.5    | 0.8854   | 541       | 70          | 130         | 71           | 15.3             | 13.3             |
> > | GLM3       | 0.9238   | 449       | 37          | 163         | 163          | 20.8             | 17.9             |

---

> > > ### Author Response · Authors · 2024-11-20
> > > **Response to Reviewer bJeQ (part 3)**
> > >
> > > ## Weakness 2.2
> > >
> > > > It would also be beneficial to include a comparison with non-LLM based approaches and a cost-benefit analysis that weighs the improved performance against potential increased costs.
> > >
> > > Take the MedHallu-ZH dataset as an example, without any hallucination detection approach, the overall accuracy of the dataset is 0.7537, including 612 accurate samples and 200 hallucinated responses that might mislead users.
> > >
> > > By incorporating the hallucination detection method, the overall accuracy can be largely improved at the cost of rejecting responses predicted to be positive. Specifically, using qwen1.5-7B-chat \+ SelfElicit, 130 over 200 hallucinated responses can be identified, leading to an overall accuracy of 0.8854 (+0.1317 points). Using ChatGLM3-8B \+ SelfElicit, 163 over 200 hallucinated responses can be identified, leading to an overall accuracy of 0.9238 (+0.1701 points).
> > >
> > > Moreover, we use another retrieval-argument baseline by: (1) using Bing search to obtain the top 5 passages related to each statement, (2) using an NLI model (StructBERT\[1\]) to predict the supportiveness, and (3) regarding unsupported statements as non-factuality. The inferior result of this baseline comes from two-fold: (i) the retrieval quality of current search engines is not good enough and (ii) besides knowledge, fact-checking also relies on reasoning. For example, a statement including a personal condition “Fasting blood glucose level of 4.2 mmol/L is normal.” requires numerical reasoning for fact-checking since the retrieval only contains general knowledge, e.g. “Normal blood glucose level for non-diabetics should be 3.9–5.5 mmol/L”.
> > >
> > > In summary, using LLM for hallucination can largely improve the overall accuracy of a QA system, at the cost of rejected responses and time cost similar to retrieval-based methods.
> > >
> > > ## Weakness 3
> > >
> > > > Potential for Induced Hallucinations: While the framework aims to mitigate hallucinations, the process of self-elicitation and reflection may still introduce inaccuracies or hallucinations, especially with ambiguous or unfamiliar content. Have the authors considered implementing additional safeguards or validation steps to catch induced hallucinations? Specific strategies or examples would be helpful in understanding how this limitation might be addressed.
> > >
> > > We acknowledge that the framework may introduce hallucinations during the reflection. However, we argue that it is inevitable for all baselines during reasoning.
> > >
> > > In this manuscript, we use conflict detection and mitigation (line 245, Section 3.3)  as safeguards, which are motivated by the self-consistency between former and latter thoughts. The ablation study (Section 4.3) shows the importance of such a safeguard mechanism in overall performance.
> > >
> > > Moreover, in Section 4.5 in the revised version, we show a case study to demonstrate that the model might generate inaccuracies for unfamiliar knowledge and how it is mitigated via conflict detection, which should provide an intuitive explanation of how the limitation be mitigated.
> > >
> > > ## Questions 1 & 3
> > >
> > > > How does the framework handle cases where the LLM lacks sufficient intrinsic knowledge about a specific topic?
> > >
> > > > How does the framework handle cases where the LLM lacks sufficient intrinsic knowledge about a specific topic? Have the authors tested the framework's performance on topics known to be outside the LLM's training data? Additionally, is there a fallback mechanism in place for when the LLM expresses low confidence?
> > >
> > > We acknowledge that the framework fails when LLM lacks sufficient intrinsic knowledge about a specific topic, which is inevitable for all baselines.
> > >
> > > Unfortunately, due to the lack of available benchmarks, we are not able to quantify the results. However, we can provide a qualitative analysis. For example, in Figure 6, Section 4.5 in the revised version, the model failed to evaluate the factuality of a statement about two medications, Dermovate and Daivonex, which are the trade names of two medications. We conclude these trade names are less familiar to the model than their pharmaceutical names by conducting manual tests. In this case, thanks to the conflict detection mechanism based on self-consistency, our framework is capable of mitigating the inaccuracies.
> > >
> > > In future works, our method can incorporate the information from external databases into the knowledge hypergraph, without modifying the framework or finetuning LLMs, providing a potential way to mitigate this limitation.

---

> ### Author Response · Authors · 2024-11-20
> **Response to Reviewer bJeQ (part 4)**
>
> ## Question 2
>
> >  What are the computational costs associated with maintaining and updating the knowledge hypergraph, especially for large datasets? Could the authors provide specific metrics or comparisons, such as how these costs scale with the size of the input or the number of iterations?
>
> Please refer to Question 3, reviewer GtBE (link: https://openreview.net/forum?id=r9mYbs8RTH&noteId=fFjI0DmHZz). The conclusion is that the complexity is $\mathcal{O}(F^2)$ where $F$ refers to the number of factoids in a sample (the number of iterations). However, since the conflict detection/mitigation procedures are not always activated and the scale of current datasets is not large enough, it is not the major issue for this task in real-world scenarios.
>
> ## Question 4
>
> > Can the approach be adapted or extended to work with external knowledge sources to further enhance hallucination detection?
>
> Yes. Conceptually, by incorporating external information into the knowledge hypergraph, our method can be enhanced without modifying the framework, which is impossible for the baselines.
>
> ## Question 5
>
> >  How does the framework perform across different domains and languages, and what adaptations might be necessary for domain-specific applications?
>
> **domain**
>
> We have conducted experiments on the WikiBio\[3\] dataset, which includes the biographies of 238 celebrities. The result shows that SelfElicit has strong performance across different domains. Detailed discussions can be found in our response to Weaknesses 8 & 11, Reviewer LceZ (link: https://openreview.net/forum?id=r9mYbs8RTH&noteId=ttwwA4UQJX)
>
> **language**
>
> Table 4: qwen1.5-7B-chat results across languages. S: sentence-level metrics. R: response-level metrics. **Bold**: better than the other language.
>
> |   Dataset   |      | SelfElicit |           |    IO     |           | ContextIO |           | HistoryIO |           |    CoT    |           |   CoVE    |           |    FaR    |           | SelfChkGPT |           | ChatProtect |           |
> | :---------: | :--: | :--------: | :-------: | :-------: | :-------: | :-------: | :-------: | :-------: | :-------: | :-------: | :-------: | :-------: | :-------: | :-------: | :-------: | :--------: | :-------: | :---------: | :-------: |
> |   Metric    |      |    *F1*    |   *AUC*   |   *F1*    |   *AUC*   |   *F1*    |   *AUC*   |   *F1*    |   *AUC*   |   *F1*    |   *AUC*   |   *F1*    |   *AUC*   |   *F1*    |   *AUC*   |    *F1*    |   *AUC*   |    *F1*     |   *AUC*   |
> | MedHallu-ZH |  S   | **0.269**  | **0.810** | **0.187** | **0.771** | **0.191** | **0.760** | **0.238** | **0.782** | **0.192** | **0.638** | **0.165** | **0.597** | **0.207** | **0.763** |   0.085    |   0.500   |  **0.085**  | **0.512** |
> |             |  R   | **0.475**  | **0.671** | **0.441** |   0.598   |   0.430   |   0.603   |   0.453   |   0.653   | **0.402** | **0.571** | **0.395** | **0.548** |   0.441   |   0.613   |   0.395    |   0.500   |    0.350    | **0.51**7 |
> | MedHallu-EN |  S   |   0.242    |   0.803   |   0.182   |   0.762   |   0.168   |   0.743   |   0.233   |   0.779   |   0.192   |   0.596   |   0.085   |   0.500   |   0.187   |   0.763   | **0.226**  | **0.682** |    0.085    |   0.505   |
> |             |  R   |   0.463    |   0.656   |   0.436   | **0.622** | **0.443** | **0.614** | **0.472** | **0.659** |   0.395   |   0.570   |   0.395   |   0.498   | **0.445** | **0.630** | **0.428**  | **0.623** |  **0.395**  | 0.50**5** |
>
> It can be observed that, although these datasets have identical knowledge points and hallucinations, the performance of MedHallu-en is relatively inferior compared with MedHallu-zh in 24 over 36 cases. Considering the overall performance is dependent on the capacity of the models, we owe that the model is unfamiliar with the English version of some medical terminologies (e.g. Chinese medicine), which has been proven by our manual tests.
>
> **adaption**
>
> The only adaption required is the prompt of extracting entities/statements (Section 3.1), which is an Open Information Extraction (OpenIE) procedure. We observe that incorporating domain-specific expertise improves the quality of the extraction. All other prompts used in our framework are domain-independent and no adaptation is required.
>
> ## References
>
> [1]StructBERT. [https://modelscope.cn/models/iic/nlp\_structbert\_nli\_chinese-large](https://modelscope.cn/models/iic/nlp_structbert_nli_chinese-large).
>
> [2]SelfCheckGPT: Zero-Resource Black-Box Hallucination Detection for Generative Large Language Models. 2023.
>
> [3]WikiBio. [https://huggingface.co/datasets/potsawee/wiki\_bio\_gpt3\_hallucination](https://huggingface.co/datasets/potsawee/wiki_bio_gpt3_hallucination)

---

### Official Review · Reviewer_LceZ · 2024-11-04

**Soundness:** 2
**Presentation:** 2
**Contribution:** 2
**Rating:** 3
**Confidence:** 4

**Summary:**

This paper introduces SelfElicit, a novel framework for detecting hallucinations in long-form content generated by Large Language Models (LLMs). The key innovation is using self-generated thoughts from prior statements to elicit a model's intrinsic knowledge, combined with a knowledge hypergraph to organize and validate the elicited information.

**Strengths:**

Pros:
1. The paper addresses a significant and timely problem in LLM research - hallucination detection in long-form content.
2. The authors demonstrate through careful ablation studies that their approach outperforms variants without self-elicitation by 3.0% on average. The elicitation quality analysis (Figure 3) provides compelling evidence that self-generated thoughts lead to both higher factuality and diversity compared to alternative approaches.
3. The authors show through their ablation studies that removing any of these components leads to performance degradation, with the full method outperforming ablated variants by 2.1-6.5% on average.

**Weaknesses:**

Cons:
1. There is no related work section. It is strongly suggested that the authors conduct a comprehensive literature review, especially on hallucination detection. Otherwise, this paper is actually not complete.
2. "For the convenience in writing, we slightly abuse the term “non-factual” to represent both factuality and faithfulness hallucinations (Huang et al., 2023). Yet our work is not limited to either one" is confusing. What type of hallucination does this paper focus on? It is suggested that the authors explicitly state the type of hallucinations rather than mix them.
3. "they are unable to learn the inherent semantic continuity in long-form content." (Line 19) is unclear. It is suggested that the authors define "semantic continuity" before they use it, even in the abstract.
4. "However, there remains a concern regarding their tendency to generate hallucinations (Bang et al., 2023), producing sentences with plausible looking yet factually1 unsupported content (Huang et al., 2023) and hurting their faithfulness in real-world scenarios expecting factually-accurate response (Wei et al., 2024)." How do the authors define "faithfulness"? It is suggested that the authors differentiate "faithfulness" and "factuality". Otherwise, it is confusing.
5. The example “Gliclazide can be taken at any time of the day→, regardless of whether it is on an empty stomach or after meals→” needs more explanations why it is misleading.
6. "While insightful, we contend that these methods either require complex manual prompts or involve intricate reasoning processes, which limit their elicitation capacity and increase the risk of accumulated inaccuracies and hallucinations." (Line 67) needs more explanations
7. The evaluated LLMs are relatively limited and small-scale. It is suggested that the authors also evaluate on SOTA models such as GPT-4 and models with a larger scale such as llama-70B. Otherwise, the effectiveness of the proposed method is relatively limited.
8. The authors only evaluate the performance of the proposed methods on their own dataset. Could the authors also test the effectiveness on public datasets?
9. The datasets are not open-sourced. The construction process of the dataset is also not clear.
10. "knowledge hypergraph" (line 87) is unclear. It is suggested that the authors define "knowledge hypergraph" before they use it.
11. The paper's evaluation is limited to the medical domain. The lack of evaluation across other domains makes it difficult to assess the general applicability of the approach.
12. The conflict resolution mechanism, while important, relies heavily on Natural Language Inference (NLI). The paper doesn't provide detailed analysis of how different NLI approaches might impact the overall performance, nor does it discuss potential failure modes of the NLI component. This is a crucial component that deserves more thorough investigation.

# Comment after Rebuttal

Thanks for the rebuttal. I decide to maintain my score and suggest another round of revision. Thanks.

**Questions:**

See weaknesses

---

> ### Author Response · Authors · 2024-11-20
> **Response to Reviewer LceZ (part 1)**
>
> We sincerely thank the reviewers for the constructive feedback.
>
> ## Weakness 1
>
> >  There is no related work section. It is strongly suggested that the authors conduct a comprehensive literature review, especially on hallucination detection. Otherwise, this paper is actually not complete.
>
> We apologize for the confusion. The related work section is included in Appendix A in our initial submission.
>
> ## Weakness 2
>
> >  "For the convenience in writing, we slightly abuse the term “non-factual” to represent both factuality and faithfulness hallucinations (Huang et al., 2023). Yet our work is not limited to either one" is confusing. What type of hallucination does this paper focus on? It is suggested that the authors explicitly state the type of hallucinations rather than mix them.
>
> This work mainly focuses on factuality (external) hallucinations and investigates how to use intrinsic knowledge to detect non-factualities. Moreover, we also observe that in several cases, our method is capable of detecting faithfulness (internal) hallucinations. We owe it to the inconsistency detection component between the former and the latter statements, which can detect internal hallucinations to some extent.
>
> We have differentiated the terminologies in line 53 in our manuscript to avoid confusion.
>
> ## Weakness 3
>
> > "they are unable to learn the inherent semantic continuity in long-form content." (Line 19\) is unclear. It is suggested that the authors define "semantic continuity" before they use it, even in the abstract.
>
> We have updated the abstract by replacing “semantic continuity” with “in-context semantics”, which is more understandable.
>
> ## Weakness 4
>
> > "However, there remains a concern regarding their tendency to generate hallucinations (Bang et al., 2023), producing sentences with plausible looking yet factually1 unsupported content (Huang et al., 2023\) and hurting their faithfulness in real-world scenarios expecting factually-accurate response (Wei et al., 2024)." How do the authors define "faithfulness"? It is suggested that the authors differentiate "faithfulness" and "factuality". Otherwise, it is confusing.
>
> We apologize for the confusion. The meaning here is that such non-factuality will weaken the **reliability** of LLMs in scenarios such as medicine. We replace the word to reliability for clearness. As stated in the above Weakness 2, this work mainly focuses on factuality (external) hallucinations and we have differentiated these terminologies in the revised version.
>
> ## Weakness 5
>
> > The example “Gliclazide can be taken at any time of the day, regardless of whether it is on an empty stomach or after meals” needs more explanations why it is misleading.
>
> “Swallow your gliclazide tablets whole with a drink of water. Do not chew them. If you are taking 2 doses a day, take 1 dose with your breakfast and 1 dose with your evening meal. If you are taking slow-release gliclazide, take your dose once a day with breakfast.” \[5\] In summary, gliclazide should be taken with the meal, rather than “at any time”. Therefore, this statement is factually incorrect. We have included a brief explanation in line 39\.

---

> ### Author Response · Authors · 2024-11-20
> **Response to Reviewer LceZ (part 2)**
>
> ## Weakness 6
>
> > "While insightful, we contend that these methods either require complex manual prompts or involve intricate reasoning processes, which limit their elicitation capacity and increase the risk of accumulated inaccuracies and hallucinations." (Line 67\) needs more explanations
>
> We analyze the limitations of existing methods as follows:
>
> * ChatProtect requires multiple reasoning steps: generate triples, elicit intrinsic knowledge via cloze, and detect hallucinations by consistency. We observe that the process of generating triples introduces most inaccuracies because some sentences include conditioned or constrained knowledge (e.g. sentence “*Gliclazide is used when diabetes cannot be controlled by proper dietary management…*” ). Simple triples discussed in ChatProtect, i.e. (sub, pred, obj), are unable to handle such cases, or careful prompt designs are required to handle such complex cases. In summary, the multi-step fact-checking reasoning process increases extra manual labor for prompt design, the cost of LLM usage, and the risk of accumulated inaccuracies.
> * SelfCheck also requires multiple reasoning steps: target extraction, information collection, step generation, and result comparison. Similarly, it has a higher cost of LLM usage and the risk of inaccuracy accumulation.
> * EVER, CoVE, and QuestGen generate fact-validation questions for each factoid, elicit intrinsic knowledge via answering these questions, and compare the results. We observe that most uncertainties and inaccuracies occur when generating these questions. For example, given a factoid *“Angina pectoris can be relieved by sublingual nitroglycerin tablets in emergency situations.”* The model sometimes generates opaque, inarticulate articulate questions “In what situations, can Angina pectoris be relieved by sublingual nitroglycerin tablets?” or open-ended answers “What medications are commonly used for relieving angina pectoris?” that might have several answers.
> * SelfCheckGPT (-prompt) and FaR ask the model to generate documents related to the factoid and check whether the factoid is supported by these documents. We observe that the generated documents tend to be lengthy and have much irrelevant information, increasing the cost and introducing noises to the evaluation reasoning.
>
> Compared with the above methods, our SelfElicit uses a simple elicitation prompt and directly expresses relevant knowledge conditioned on the evaluation result, which reduces the overhead of LLM usage and the risk of inaccuracy accumulation. Experimentally, these baselines use 76%-1244% more token generations (Table 2\) while having inferior performance (Table 1). We will elaborate on their limitations more clearly in the revised manuscript.
>
> Table 1: Qwen2.5, Llama3.1, and GPT4o-mini results on MedHallu-EN. S: sentence-level metrics. R: response-level metrics. Bold: the best. Italic: the second best.
>
> |  Methods   |      | SelfElicit |           |   IO    |         | ContextIO |         | HistoryIO |         |    CoT    |         |  CoVE   |           |  FaR  |       | SelfChkGPT |       | ChatProtect |       |
> | :--------: | :--: | :--------: | :-------: | :-----: | :-----: | :-------: | :-----: | :-------: | :-----: | :-------: | :-----: | :-----: | :-------: | :---: | :---: | :--------: | :---: | :---------: | :---: |
> |   Metric   |      |    *F1*    |   *AUC*   |  *F1*   |  *AUC*  |   *F1*    |  *AUC*  |   *F1*    |  *AUC*  |   *F1*    |  *AUC*  |  *F1*   |   *AUC*   | *F1*  | *AUC* |    *F1*    | *AUC* |    *F1*     | *AUC* |
> |   Qwen2    |  S   | **0.282**  | **0.820** | *0.275* |  0.805  |   0.247   |  0.802  |   0.254   | *0.811* |   0.211   |  0.636  |  0.259  |   0.672   | 0.217 | 0.784 |   0.232    | 0.675 |    0.087    | 0.523 |
> |            |  R   | **0.479**  | **0.667** | *0.466* | *0.665* |   0.460   |  0.661  |   0.456   |  0.656  |   0.422   |  0.422  |  0.440  |   0.614   | 0.447 | 0.640 |   0.444    | 0.636 |    0.395    | 0.537 |
> |   Llama3   |  S   |  *0.211*   | **0.773** |  0.156  |  0.724  |   0.170   | *0.741* |   0.147   |  0.662  | **0.223** |  0.666  |  0.184  |   0.699   | 0.184 | 0.730 |   0.158    | 0.634 |    0.208    | 0.601 |
> |            |  R   |  *0.447*   |  *0.622*  |  0.406  |  0.546  |   0.405   |  0.572  |   0.413   |  0.605  | **0.449** |  0.626  |  0.421  |   0.562   | 0.422 | 0.586 |   0.417    | 0.613 |    0.414    | 0.600 |
> | GPT4o mini |  S   | **0.329**  |   0.682   |  0.185  |  0.560  |   0.183   |  0.564  |   0.250   |  0.597  |  *0.279*  | *0.686* |  0.277  | **0.703** | 0.085 | 0.520 |   0.135    | 0.623 |    0.085    | 0.512 |
> |            |  R   | **0.494**  | **0.668** |  0.395  |  0.559  |   0.395   |  0.574  |   0.395   |  0.586  |   0.487   |  0.661  | *0.488* |  *0.658*  | 0.395 | 0.521 |   0.395    | 0.603 |    0.395    | 0.505 |

---

> ### Author Response · Authors · 2024-11-20
> **Response to Reviewer LceZ (part 3)**
>
> ## Weakness 7
>
> > The evaluated LLMs are relatively limited and small-scale. It is suggested that the authors also evaluate on SOTA models such as GPT-4 and models with a larger scale such as llama-70B. Otherwise, the effectiveness of the proposed method is relatively limited.
>
> **Model Scale**
>
> We have shown the scalability results using up to 110B model (Qwen1.5-110B) in Figure 5, Appendix A6 in the initial submission (or Figure 7, Appendix D1 in the revised version). The scalability results have demonstrated that (i) larger models tend to have better performance, (ii) the inference costs also increase nearly linearly with the model size, and (iii) our SelfElict consistently outperforms the baselines.
>
> **Comparison with other SOTAs**
>
> We have also conducted further experiments on more LLMs, including SOTA open-sourced models Qwen2.5, Llama3.1, and commercial model GPT4o-mini. The results are shown in Table 1.
>
> The results show that our SelfElict outperforms all baselines in most cases across the latest open-source LLMs and commercial LLMs, which demonstrates the versatility of our proposed approach. Moreover, we can also observe that the latest-generation models (Qwen2.5 and Llama3.1) outperform their previous-generation counterparts (Qwen1.5 and Llama2). The full results are shown in Table 1 in our revised version.
>
> We also notice that GPT4o-mini, one of the most efficient and powerful commercial models, has relatively inferior performance. This reason is that *we are not able to obtain the model’s confidence score nor adaptively adjust the thresholds to reduce confidence biases.* Specifically, when it comes to commercial models accessed by APIs, we are not able to obtain detailed token logits but only binary results through verbal outputs. Consequently, it is unable to adaptively modify the thresholds according to the results of the validation set. The confidence biases of all methods might eventually lead to sub-optimal performance\[2\]. Moreover, since the training corpora of GPT are not public, we can not further assess whether it has specific knowledge in the medical domain.
>
> ## Weaknesses 8 & 11
>
> > The authors only evaluate the performance of the proposed methods on their own dataset. Could the authors also test the effectiveness on public datasets?
>
> > The paper's evaluation is limited to the medical domain. The lack of evaluation across other domains makes it difficult to assess the general applicability of the approach.
>
> We acknowledge that it is important to comprehensively evaluate the methods with public datasets in various domains.
>
> We have added an experiment on a publicly available long-form hallucination dataset, WikiBio\[1\], which includes the biographies of 238 celebrities. Since most samples in this dataset are positive (hallucination), we only report the AUC metric here.
>
> Table 2: Results on WikiBio. S: sentence-level metrics. R: response-level metrics. Bold: the best. Italic: the second best.
>
> | Methods |      | avg improv. | SelfElicit |  IO   | ContextIO | HistoryIO |  CoT  | CoVE  |  FaR  | SelfChkGPT | ChatProtect |
> | :-----: | :--: | :---------: | :--------: | :---: | :-------: | :-------: | :---: | :---: | :---: | :--------: | :---------: |
> | Metric  |      |             |   *AUC*    | *AUC* |   *AUC*   |   *AUC*   | *AUC* | *AUC* | *AUC* |   *AUC*    |    *AUC*    |
> |  Qwen   |  S   |    11.1%    | **0.594**  | 0.527 |  *0.587*  |   0.543   | 0.500 | 0.527 | 0.543 |   0.539    |    0.512    |
> |         |  R   |    12.2%    |  *0.653*   | 0.628 |   0.522   |   0.614   | 0.566 | 0.524 | 0.508 |   0.639    |  **0.657**  |
> | Llama2  |  S   |    5.6%     |  *0.556*   | 0.516 |   0.534   |   0.477   | 0.534 | 0.553 | 0.506 | **0.572**  |    0.517    |
> |         |  R   |    18.0%    |  0.698\*   | 0.559 |   0.534   |   0.540   | 0.531 | 0.636 | 0.522 | **0.708**  |   *0.704*   |
>
> Although SelfElicit does not achieve the best performance in the starred (\*) metric, the difference is less than 2%. Compared against baselines, our SelfElicit achieves leading performance (best or second best in 3/4 cases) and on average improvements of 11.1%, 12.1%, 5.6%, and 18.0% against all baselines. These results show that SelfElicit has strong performance across different domains.
>
> Moreover, we also observe that the relative performance is not as obvious as on the MedHallu datasets. The reasoning might be that the biographies have weaker contextual relationships between sentences but features stating individual factoids, which can be proven by the inferior performance of context-argument methods (ContextIO and HistoryIO).

---

> > ### Author Response · Authors · 2024-11-20
> > **Response to Reviewer LceZ (part 4)**
> >
> > ## Weakness 9
> >
> > > The datasets are not open-sourced. The construction process of the dataset is also not clear.
> >
> > The dataset is fully accessible in our public anonymized repository: https://anonymous.4open.science/r/SelfElicit-DFCE.
> >
> > The construction process of the datasets is shown in Appendix A4 in the initial submission (or Appendix D2 in the revised version).
> >
> > ## Weakness 10
> >
> > > "knowledge hypergraph" (line 87\) is unclear. It is suggested that the authors define "knowledge hypergraph" before they use it.
> >
> > We apologize for the confusion and have updated the corresponding statements.
> >
> > ## Weakness 12
> >
> > > The conflict resolution mechanism, while important, relies heavily on Natural Language Inference (NLI). The paper doesn't provide detailed analysis of how different NLI approaches might impact the overall performance, nor does it discuss potential failure modes of the NLI component. This is a crucial component that deserves more thorough investigation.
> >
> > In Appendix A11 in the initial submission (or Appenidx E3 in the revised version), we have compared LLM-based and specialized pre-trained NLI methods (StructBERT\[3\] for MedHallu-zh and DeBERTa\[4\] for MedHallu-en). The results show that the differences between them are trivial and therefore we decided to use prompts in our implementation in the main text for convenience.
> >
> > We acknowledge that there exist some potential failure modes of the NLI component. For example, (i) since the NLI component is activated only if the new reflections $e^{new}$ and the original statements $e^{orig}$ in the graph share an identical vertice set (line 253), in cases when the generated reflection shares no vertices with existing edges, the NLI component will not be activated. (ii) Another case is that the NLI method fails to correctly identify semantic relationships. In both examples, the hallucinated reflections will be retained in the graph. However, despite these limitations, the NLI component still plays an important role in the overall performance of SelfElict (Section 4.4 Ablation Study).
> >
> > ## References
> >
> > \[1\]WikiBio. [https://huggingface.co/datasets/potsawee/wiki\_bio\_gpt3\_hallucination](https://huggingface.co/datasets/potsawee/wiki_bio_gpt3_hallucination)
> >
> > \[2\]Uncertainty in Language Models: Assessment through Rank-Calibration. 2024\.
> >
> > \[3\]StructBERT. [https://modelscope.cn/models/iic/nlp\_structbert\_nli\_chinese-large](https://modelscope.cn/models/iic/nlp_structbert_nli_chinese-large)
> >
> > \[4\]DeBERTa. [https://huggingface.co/microsoft/deberta-large-mnli](https://huggingface.co/microsoft/deberta-large-mnli)
> >
> > \[5\]https://www.nhs.uk/medicines/gliclazide/how-and-when-to-take-gliclazide/

---

### Author Response · Authors · 2024-11-20
**Response to AC and all reviewers**

We sincerely thank the reviewers for thoroughly examining this manuscript and providing valuable insights for the further improvements of this work.

We have carefully considered these important suggestions and revised them accordingly. Please find our detailed responses to the reviewers' comments below. Additionally, we have made the following improvements to the paper (important modifications are highlighted in the main text in our revised version). Some important modifications are listed as follows:

+ Additional experiments on **a public dataset about biography**, namely WikiBio, are included. The results show that our method generates well across domains.
+ Additional experiments with **state-of-the-art modern LLMs** including Qwen2.5, Llama3.1, and gpt4o-mini is included.  The results show that our method has superior performance across LLM backbones.
+ Re-categorized the baselines to highlight their characteristics with various elicitation approaches.
+ **Add two context-argument baselines** (ContextIO and HistoryIO) for a more intuitive comparison. We have also modified Table 1 to show more metrics.
+ Two **case studies** for qualitative analysis are included.
+ Clarify the main focus of this paper is the factual (external) hallucination and carefully differentiate the terminologies.
+ Redesign Figure 2 for a better understanding of the overall framework.
+ The Appendix is included at the end of the main text rather than a separate file with modifications.

---

### Note · Authors · 2024-12-05

I have read and agree with the venue's withdrawal policy on behalf of myself and my co-authors.